# Drifting snow statistics from multiple-year autonomous measurements in Adelie Land, eastern Antarctica

Charles Amory

Department of Geography, University of Liège, Liège, Belgium

**Correspondence:** C. Amory (charles.amory@uliege.be)

**Abstract.** Drifting snow is a widespread feature over the Antarctic ice sheet whose climatological and hydrological significances at the continental scale have been consequently investigated through modelling and satellite approaches. While field measurements are needed to evaluate and interpret model and satellite products, most drifting snow observation campaigns in Antarctica involved data collected at a single location and over short time periods. With the aim of acquiring new data relevant to the observation and modelling of drifting snow in Antarctic conditions, two remote locations in coastal Adelie Land (East Antarctica) 100 km apart were instrumented in January 2010 with meteorological and second-generation IAV Engineering acoustic FlowCapt™ sensors. The data, provided nearly continuously so far, constitutes the longest dataset of autonomous near-surface (i.e. within 2 m) measurements of drifting snow currently available over the Antarctic continent. This paper presents an assessment of drifting snow occurrences and snow mass transport from up to 9 years (2010-2018) of half-hourly observational records collected in one of the Antarctic regions most prone to snow transport by wind. The dataset is freely available to the scientific community and can be used to complement satellite products and evaluate snow-transport models close to the surface and at high temporal frequency

## 1 Introduction

Wind-driven transport of snow in Antarctica, organized in drifting (< 2 m above ground level) and blowing (> 2 m above ground level) snow, has important implications for the ice-sheet climate and surface mass balance. Erosive winds redistribute snow at the surface and can form areas of near-zero net accumulation (known as wind glaze areas) or even net ablation (known as blue ice areas) whose presence has a profound influence on the local surface energy balance (Bintanja, 1999; Scambos et al., 2012), possibly enhancing surface melt (Lenaerts et al., 2017). In coastal areas, wind redistribution of snow is responsible for an export of mass beyond the ice-sheet margins (Scarchilli et al., 2010; Palm et al., 2017). Sublimation of snow particles during transport is a major component of the surface heat and moisture budgets in regions where most of the precipitated snow is relocated by wind (e.g., Mann et al., 2000; Bintanja, 2001; Thiery et al., 2012).

Because of the widespread character of drifting and blowing snow over the vast and remote Antarctic continent, estimates of their hydrological and climatological significances at the ice-sheet scale rely on parameterized methods (e.g., Gallée, 1998; Déry and Yau, 2002; Lenaerts and van den Broeke, 2012; Palm et al., 2017; van Wessem et al., 2018; Agosta et al., 2019). A consensus emerging from these efforts that has persisted for more than two decades suggests that, although significant locally,

mass loss through wind redistribution and export into the ocean is of minor importance while sublimation during transport remains the dominant sink of mass when evaluated over the whole ice sheet. Conversely, contrasting results can be found from one study to another in the absolute values attributed to the contribution of wind-driven snow processes to the large-scale mass transport. Latest continent-wide estimations of wind-driven snow sublimation obtained from regional modelling (van Wessem et al., 2018) are lower by a factor of 4 than those computed from a combination of satellite products and meteorological reanalysis (Palm et al., 2017). Modelled snow mass fluxes presented in Agosta et al. (2019) exhibit a similar overall spatial pattern but are more than 3 times larger than those reported in van Wessem et al. (2018). Considering the diversity of interactions and the non-linearity of processes involved in the onset, development and magnitude of wind-driven snow occurrences (e.g., Déry et al., 1998; Bintanja, 2000; Amory et al., 2016), model results as well as the assumptions made in the implementation of wind-driven snow physics need to be carefully assessed with independent observations.

Advances in active lidar remote sensing of the atmosphere from space have provided recent insights into the spatial distribution and temporal variability of blowing snow over the last decade independently from modelling approaches. Although of unrivalled interest for studying blowing snow over large temporal, horizontal and vertical scales simultaneously, satellite lidar data provide snapshots of a particular set of blowing snow properties (frequency, layer depth, optical thickness) relative to the satellite revisit time (Palm et al., 2011). Moreover, prior satellite detection has been restricted to clear-sky or optically thin cloud conditions and relatively deep (> 30 m) blowing snow layers, precluding its application for characterization of shallower (drifting and blowing snow) layers and for model evaluation in the vicinity of the surface. While this last limitation is also shared with ground-based remote sensing techniques (Mahesh et al., 2003; Gossart et al., 2017), measured vertical profiles of snow mass fluxes display however the strongest gradients in the lowest metres of the atmosphere (Budd, 1966; Mann et al., 2000; Nishimura and Nemoto, 2005).

Direct near-surface observations of wind-driven snow in Antarctica are sparse in time and space to the extent that long-term quality-controlled datasets that yet constitute essential development and evaluation bases for parametrization schemes barely exist. The absence of an official standard instrument has led to the use of a wide range of observation techniques from mechanical traps and nets to electronic (optical, piezoelectric, acoustic) sensors (see Leonard et al. (2012) and Trouvilliez et al. (2014) for an extensive review) as well as visual observations carried out at some Antarctic manned stations (Mahesh et al., 2003; König-Langlo and Loose, 2007). However, like satellite products, visual observations are representative of instantaneous conditions only and are additionally dependent on personal appreciation of the observer who might change with time, leading to non-uniform and temporally discontinuous records.

In spite of their disparity, near-surface measurements of wind-driven snow over the Antarctic ice sheet have provided valuable and accurate information that cannot be sensed remotely nor determined visually. This includes, among others, particle size distributions and related dimensionless shape parameters, total particle numbers and snow mass fluxes at different heights. Although the data collected are also relative to the instrument used and can hardly compare to each other, they are eventually useful for modelling experiments. The dimensionless shape parameter and particle number are, for instance, either predicted or prescribed quantities in snow-transport models that compute sublimation rates and snow mass fluxes assuming a gamma distribution of particles (e.g., Déry et al., 1998; Déry and Yau, 1999, 2001; Bintanja, 2000; Nemoto, 2004; Lenaerts et al.,

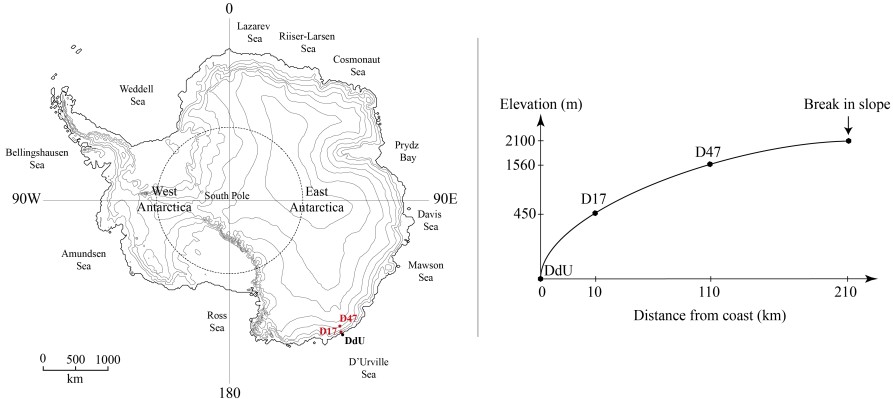

**Figure 1.** Location of Dumont d'Urville station (DdU) and sites D17 and D47 in coastal Adelie Land and schematic cross-section showing elevation and distance from the coastline for each site. Contours show elevation each 500 m from 0 to 4,500 m.

2012). Additionally observed snow mass fluxes can be directly used to assess the ability of models to reproduce wind-driven snow conditions at a specific location in a qualitative (e.g., Lenaerts et al., 2012; Gallée et al., 2013) or a quantitative (e.g., Nishimura and Nemoto, 2005; Yang and Yau, 2007; Amory et al., 2015; van Wessem et al., 2018) perspective. However, even if each dataset is individually valuable regarding the scarcity of observations, in most cases the data were collected at a single location and over a few months, precluding investigations into spatial and temporal (seasonal and interannual) variability.

In order to acquire new model-evaluation oriented observations, a field campaign specifically dedicated to drifting snow has been initiated in January 2010 in Adelie Land (Trouvilliez et al., 2014), a wind confluence area of East Antarctica. Two distinct locations, namely D17 and D47 (Fig. S1), were instrumented for long-term data acquisition and equipped with second-generation IAV Engineering acoustic FlowCapt™ sensors[1] (hereafter referred to as 2G-FlowCapt™), which are particularly well-suited for continuous monitoring in remote environments and under harsh conditions (Trouvilliez et al., 2015). This study presents an assessment of drifting snow occurrences and snow mass transport from analysis of multiple-year timeseries of meteorological data and snow mass fluxes collected in this framework within a katabatic wind region of the Antarctic ice sheet among the most prone to snow transport by wind.

## 2 Site characteristics and data

### 2.1 Instrumentation

The study area consists of a sloping snowfield with a break-in-slope at nearly 210 km inland at about 2,100 m a.s.l., downstream of which D47 and D17 are located (Fig. 1). The two measurement sites are 100 km apart, south-west of the permanent French station Dumont d'Urville (66.6°S, 140°W, 40 m a.s.l.). Because of their remote locations, access and maintenance activities

---

[1]http://www.flowcapt.com

**Table 1.** Geographical and climate characteristics of the two measurement locations for the respective observation periods.

| Station | D47[a] | D17[b] |
|---|---|---|
| Start of observation | 9 Jan. 2010 | 3 Feb. 2010 |
| End of observation | 27 Dec. 2012 | 31 Dec. 2018[c] |
| Location | 67.4°S, 138.7°E | 66.7°S, 139.9°E |
| Altitude | 1,560 | 450 |
| Distance from coast (km) | 110 | 10 |
| Wind speed (m s$^{-1}$) | 11.9 | 9.8 |
| Air temperature (°C) | -25.1 | -15.5 |
| Air relative humidity[d] (%) | 90.6 | 81.4 |
| Wind direction (deg) | 158 | 154 |
| Directional constancy | 0.95 | 0.92 |

[a] Mean values at sensor level are used.

[b] Mean values at sensor level nearest to 2 m are used.

[c] The station at site D17 is still operative.

[d] With respect to ice.

are only possible in summer. At D17, a 7-m high mast is equipped with six levels logarithmically spaced (initial heights of 0.8,
1.3, 2, 2.8, 3.9 and 5.5 m) of anemometers and thermo-hygrometers housed in naturally ventilated MET21 radiation shields
(Fig. S1, left panel). The meteorological mast is oriented toward the prevailing wind direction to prevent flow distortion by
the measurement structure. The wind direction is sampled at the upper level only. Site D47 is equipped with only one level
of wind speed and direction measured at 2.8 m and temperature and relative humidity measured at 2.2 m (Fig. S1, right
panel). The thermo-hygrometers are factory calibrated to report relative humidity with respect to liquid water. Goff and Gratch
(1945) formulae are used to convert to relative humidity with respect to ice for air temperatures below 0 °C, using the sensor
temperature reports in the conversion. Ultrasonic depth gauges are used to monitor surface height changes at both sites, from
which the elevation of the sensors above the surface is assessed throughout the year. At D17, this information is not available
before December 2012 when the height ranger was deployed. The station is currently still operative, and the instruments along
the profile are raised back manually to original heights at the beginning of each summer field campaign. The remoteness and
the frequently harsh weather conditions of D47 allowed for limited servicing time, so that summer visits were restricted to the
maintenance of sensors without raising operations. As a result the measurement heights decreased from their initial values to
respectively 1.5 m for wind speed and direction and 0.9 m for temperature and relative humidity in late December 2012 when
the equipment was entirely removed. The instrument types and specificities are summarised in Table S1. Data were sampled at
15-s intervals, and stored at a half-hourly time resolution on a Campbell CR3000 datalogger.

## 2.2 Climate settings

The surface climate in coastal Adelie Land is dominated by intense, frequent and persistent katabatic flows originating from the continental interior where strong temperature inversions develop. The local topography controls the drainage of the cold, dense near-surface air as it flows downslope and accelerates toward the steep coastal escarpment over an unobstructed snow-covered fetch of several hundreds of kilometres. Table 1 lists geographical settings and climate information for the two sites. Wind speed and temperature regimes at 2-m height at the two measurement locations follow an annual cycle typical of katabatic wind confluence areas (Fig. S2). Lower temperatures and higher wind speeds are observed in winter as a result of the strong radiative deficit of the surface and increased katabatic forcing. In summer, the absorption of shortwave radiation by the surface diminishes the katabatic forcing, air temperature increases and wind speed reduces. The higher incidence of drifting snow (Fig. S3) and inherent loading of air masses with moisture through sublimation (Amory and Kittel, 2019) combined with lower temperatures in winter account for an increase in near-surface relative humidity compared to summer values. Substantially lower temperatures and subsequent dampened seasonal variations in relative humidity are observed at D47 due to the higher elevation.

Even if D17 is located near the downstream end of the sloping ice terrain where stronger katabatic forcing can be expected, year-round higher wind speeds are consistently observed at D47 some 100 km inland, as already reported by Wendler et al. (1993). Although the question remains open for further study, an explanation for this feature may involve the deceleration and subsequent thickening of the atmospheric boundary layer flow beyond the ice-sheet margins where it is no longer sustained by the buoyancy (katabatic) force. The resulting accumulation of cold air downstream over the ocean leads to the establishment of an upslope pressure gradient force opposing the katabatic flow that is responsible for an additional slowing of the airstream when reaching the coastal area (Gallée and Pettré, 1998), possibly accounting for the lower wind speeds at D17 compared to D47.

Both measurement sites show a very high constancy in wind direction (defined as the ratio of the resultant wind speed to the mean wind speed), reflecting the quasi-unidirectional nature of the flow in coastal Adelie Land (Table 1; Fig. S1). This provides evidence that topographic channelling strongly controls the surface wind regime, and indicates that cyclonic disturbances do not significantly alter the direction of the main flow.

## 2.3 Drifting snow data

### 2.3.1 Measurement principle

At each station the meteorological records were complemented by drifting snow measurements made with 2G-FlowCapt™ sensors. The instrument consists of a 1 m long tube containing electroacoustic transducers that measure the acoustic vibration caused by the impacts of windborne snow particles on the tube. Using spectral analysis, the sensor accurately distinguishes the low-frequency noise generated by turbulence from the high-frequency drifting snow signal, which is proportional to the snow mass flux integrated over the length of the tube (Chritin et al., 1999). This means that the measured acoustic vibration, and thus, the estimation of the snow mass flux depends on the shape, size, density and speed of each individual particle colliding

with the tube (Cierco et al., 2007). As precipitating snow particles directly originating from clouds and drifting (saltating and/or suspended) snow particles relocated from the ground cannot be discriminated, measured snow mass fluxes account for all forms of wind-driven snow along the sampling height. The 2G-FlowCapt™ can record continuous information as long as it remains partially exposed. This is an advantage over visual observations and satellite products provided at sporadic intervals. Moreover, the ability of these sensors to detect events of small magnitude is particularly interesting, as remote sensing techniques can only retrieve information on blowing snow layers for which the snow particles are lifted at several tens of metres off the surface (Mahesh et al., 2003; Palm et al., 2011; Gossart et al., 2017).

### 2.3.2 Field installation

In early January 2010 at D47, two 2G-FlowCapt™ were installed and superimposed vertically, with the bottom of the lower sensor located close to the surface (~0.1 m) in order to detect the onset of drifting snow (Fig. S1). At D17 two sensors were deployed in February 2010 but only one was initially installed close to the surface while the other one was set up at the top of the measurement structure. The upper sensor was removed in January 2011 because of malfunction, and reinstalled after repair in late December 2012 similarly to the configuration adopted for D47. Like for the other meteorological instruments, the 2G-FlowCapt™ sensors at D17 were reset into their original position with the lower sensor near the surface during each summer visits, except for austral summers 2015-2016 and 2016-2017 during which the pair of instruments was left unchanged. Consequently, substantial burial of the lower sensor took place during the 3-year period from early 2015 to late 2017 depending on snow accumulation and ablation. As no raising operations were undertaken at D47 the measurement structure progressively buried and the lower 2G-FlowCapt™ became entirely covered with snow during the course of the year 2012. The FlowCapt™ is low-power consuming and designed to withstand harsh climate conditions without regular human attendance. At each station battery voltage is monitored and stored together with the meteorological variables in the datalogger to ensure that the entire measurement system is sufficiently supplied with energy throughout the winter. The 2G-FlowCapt™ are continuously solicited by the datalogger (RS232 connection), such that instances of instrument malfunction (absence of response and no data) can be unambiguously distinguished from the absence of drifting snow (data containing null values). A thorough check on the observations was performed and resulted in omission of misleading data wherever necessary. Except for those very few cases, maintenance periods in summer and a major 2-month failure of the lower 2G-FlowCapt™ sensor at D47 in May and June 2012, the dataset is continuous along the respective measurement periods.

### 2.3.3 Accuracy assessment

While FlowCapts™ sensors can detect the occurrence of snow transport with a high level of confidence, the ability of the original design to estimate snow mass fluxes is more questionable (Cierco et al., 2007). These accuracy issues, without being necessarily solved, have been significantly improved with the 2G-FlowCapt™, facilitating its use for quantitative applications (Trouvilliez et al., 2015). Although measurement uncertainty is not known, the 2G-FlowCapt™ was shown to generally underestimate the snow mass flux relative to integrated estimates computed from optical measurements made with a snow particle counter S7 (SPC-S7; taken as a reference in the study) during a winter season in the French Alps, particularly during concurrent

precipitation (Trouvilliez et al., 2015). During mixed drifting snow events when erosion occurs simultaneously with snowfall, the density of precipitating particles which have not yet reached the ground is lower than eroded, more rounded snow particles originating from the ground which have lost their original crystal shape and size through collision, sublimation and the thermal processes of metamorphism. For a given snow mass flux, the particles' momentum, and by extension the measured acoustic pressure, is therefore lower during a mixed drifting snow event than during an event predominantly driven by the erosion process. This results in an underestimation of the snow mass flux measured by the 2G-FlowCapt™ during mixed events, with a magnitude depending on the relative proportion of eroded particles against fresh snow particles.

Environmental conditions influence greatly the estimation of the snow mass flux by the 2G-FlowCapt™. The intercomparison experiment in the Alps was done within a range of mass flux values ($< 2.5 \ 10^{-2}$ kg m$^{-2}$ s$^{-1}$) significantly lower than those encountered in Adelie Land (see Sect. 3.4). In addition, comparatively stronger surface winds and lower temperature on the Antarctic ice sheet favor the breaking and rounding of snow particles. This suggests that the performance of the 2G-FlowCapt™ remains to be assessed in the extreme Antarctic environment, in which large proportions of small, rounded particles can be expected in drift conditions (i.e. within 2 m above ground) even with concurrent precipitating snow (Nishimura and Nemoto, 2005).

A field experiment involving measurements with SPC-S7 and 2G-FlowCaptTM sensors performed during a 24 hour long snow transport event was undertaken at site D17 in late January 2014 (Fig. S3). Strong drift conditions were observed with 2-m wind speeds and snow mass fluxes reaching up to 19 m s$^{-1}$ and 4 $10^{-1}$ kg m$^{-2}$ s$^{-1}$ respectively. Although the statistical representativeness of the results may be small due to the low amount of data collected during only one event, the comparison shows that the snow mass fluxes provided by the two types of sensors are very similar in magnitude (Fig. S4). Further details on the experimental set-up and comparison methodology are provided in supplementary materials (Sect. S1).

### 2.3.4 Computation of drifting snow frequency and mass transport

To remove electronic or turbulence noise and ensure that actual occurrences are detected, drifting snow has been considered to occur when the half-hourly mean of the snow mass flux exceeds a confidence threshold of $10^{-3}$ kg m$^{-2}$ s$^{-1}$ as determined from visual observations on the field in Adelie Land (Amory et al., 2017). Note that the same confidence threshold yielded a high level of agreement (98.6 %) between the SPC-S7 and 2G-FlowCapt™ in terms of occurrence detection in the comparison study led by (Trouvilliez et al., 2015) in the Alps. Since this value remains small compared to snow mass fluxes estimated during drifting snow occurrences (see Sect. 3.4), the confidence threshold is assumed independent on the exposed length of the sensor. The sensor is considered unburied as long as at least 10 % (i.e. 0.1 m) of its initial length remain uncovered with snow.

Changes in the exposed length of the 2G-FlowCapt™ through snow accumulation and ablation affect the estimation of the snow mass flux as it is vertically integrated over the uncovered part of the instrument. This is a matter of concern at both sites since precipitation along the Adelie coast occurs year round almost exclusively in the form of snowfall with a mean accumulation amounting to 362 mm water equivalent per year (Agosta et al., 2012) and frequent, high wind speeds induce frequent erosion/deposition of snow. As a result, the actual sampling height varied substantially and non-uniformly throughout the measurement period preventing direct comparisons of snow transport amounts over time. This is accounted for in a simple

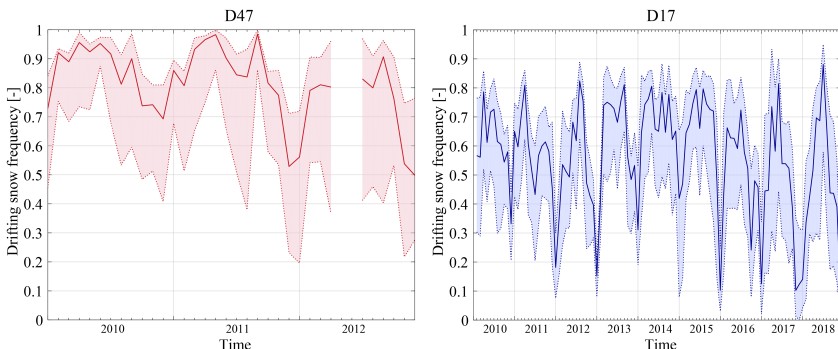

**Figure 2.** Seasonal variability of drifting snow frequency as recovered by the 2G-Flowcapt™ instruments. Shaded areas correspond to frequencies respectively computed using a relaxed and a stricter confidence threshold of $10^{-4}$ kg m$^{-2}$ s$^{-1}$ and $10^{-2}$ kg m$^{-2}$ s$^{-1}$ and are shown as a measure of uncertainty. The absence of data at D47 during May and June 2012 is due to instrument malfunction.

way by combining, when available, half-hourly snow mass fluxes from the two measurement levels to derive a standardized estimate of the drifting snow mass flux (i.e. vertically integrated between 0 and 2 m over the snow surface) $\eta_{DR}$ such that

$$\eta_{DR} = \begin{cases} \eta_1 + \eta_2, & h_1 + h_2 \geqslant h_{ref} \\ \eta_1 + \eta_2 \cdot \frac{h_{ref}}{h_1 + h_2} & h_1 + h_2 < h_{ref} \end{cases} \tag{1}$$

where $\eta_i$ (kg m$^{-2}$ s$^{-1}$) is the observed snow mass flux integrated over the exposed height $h_i$ (m) of the corresponding 2G-FlowCapt™ sensor, and $h_{ref}$ = 2 m corresponds to the sum of two fully exposed 1 m long 2G-FlowCapt™ sensors. In other words, when $h_1 + h_2 < 2$ m, it is assumed that the measured snow mass flux is constant up to 2 m. To keep consistency with the confidence threshold for the detection of drifting snow occurrences, snow mass fluxes below $10^{-3}$ kg m$^{-2}$ s$^{-1}$ have been set to zero. The horizontal snow mass transport in drift conditions for a given period of time $[t_0, t_n]$, $Q_{DR}$, then writes

$$Q_{DR}(t) = \int_{t_0}^{t_n} \eta_{DR}(t) \, dt \tag{2}$$

## 3 Analysis of observations

### 3.1 Spatial and temporal variations in drifting snow occurrences

Monthly values of drifting snow frequency at D47 and D17 indicate that drifting snow is a regular feature of the coastal slopes of Adelie Land (Fig. 2; overall averages of 0.81 at D47 and 0.57 at D17). Frequency values have been computed for each month of the observation period as the ratio between the number of half-hourly observations with a snow mass flux at the lower, unburied level $\eta_i$ higher than the confidence threshold of $10^{-3}$ kg m$^{-2}$ s$^{-1}$ and the total number of observations in that month. On each panel the shaded area corresponds to the frequency respectively computed using a relaxed and a stricter threshold of

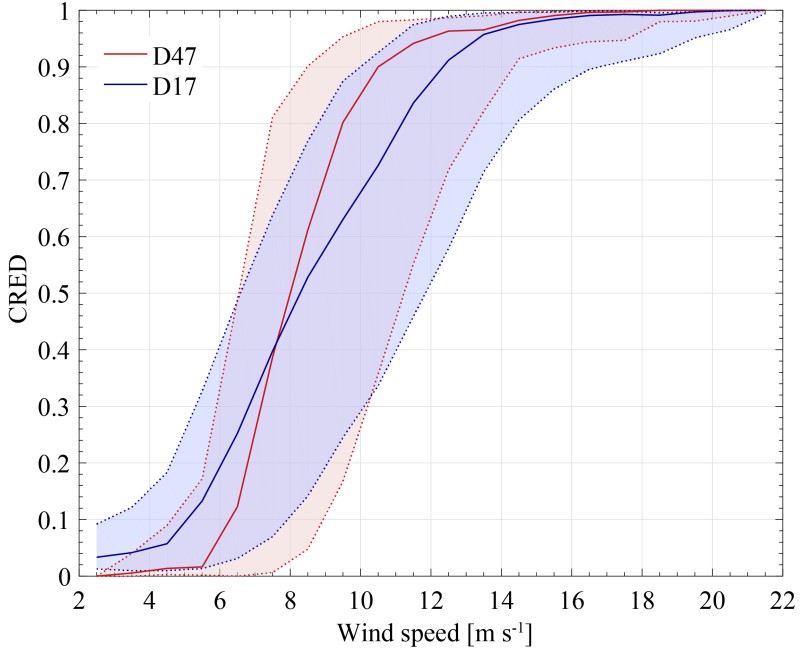

**Figure 3.** CRED distribution showing the increasing probability of observing drifting snow with increasing 2-m wind speed at sites D47 (red curve) and D17 (blue curve). Shaded areas correspond to CREDs respectively computed using a relaxed and a stricter confidence threshold of $10^{-4}$ kg m$^{-2}$ s$^{-1}$ and $10^{-2}$ kg m$^{-2}$ s$^{-1}$ and are shown as a measure of uncertainty.

**Table 2.** Standardized estimates of annual horizontal snow mass transport in drift conditions.

| Year | Snow mass transport [kg m$^{-2}$] | |
| --- | --- | --- |
| | D17 | D47 |
| 2010 | - | 1.89 $10^6$ |
| 2011 | - | 1.64 $10^6$ |
| 2012 | - | - |
| 2013 | 2.05 $10^6$ | - |
| 2014 | 2.42 $10^6$ | - |
| 2015 | 2.68 $10^6$ | - |
| 2016 | 2.63 $10^6$ | - |
| 2017 | 2.12 $10^6$ | - |
| 2018 | 2.20 $10^6$ | - |

$10^{-4}$ kg m$^{-2}$ s$^{-1}$ and $10^{-2}$ kg m$^{-2}$ s$^{-1}$ and is shown as a measure of uncertainty. While no particular inter-annual variability

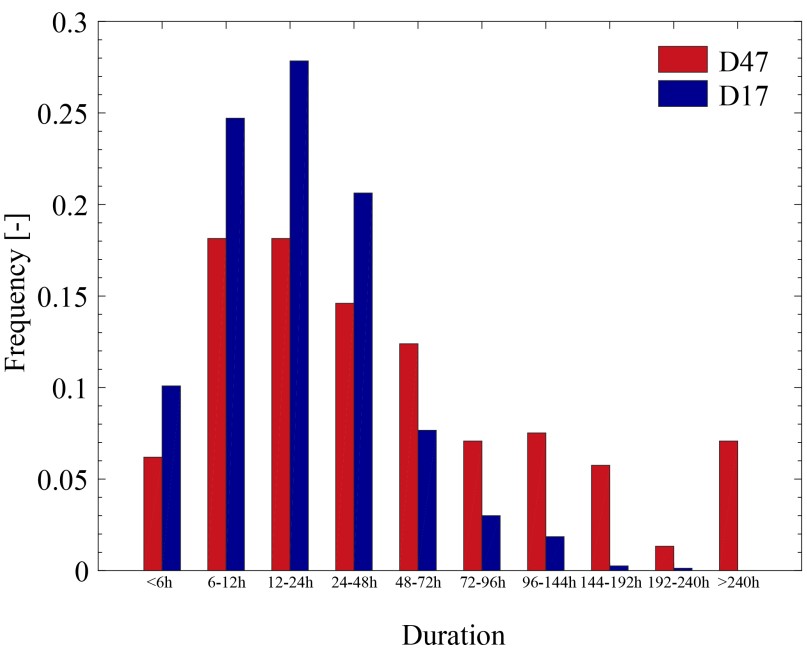

**Figure 4.** Distribution of durations of drifting snow events at D47 (red) and D17 (blue). The minimum values of duration and drifting snow mass transport for an event to be retained in the statistics are respectively set to 4 hours and 15 kg m$^{-2}$.

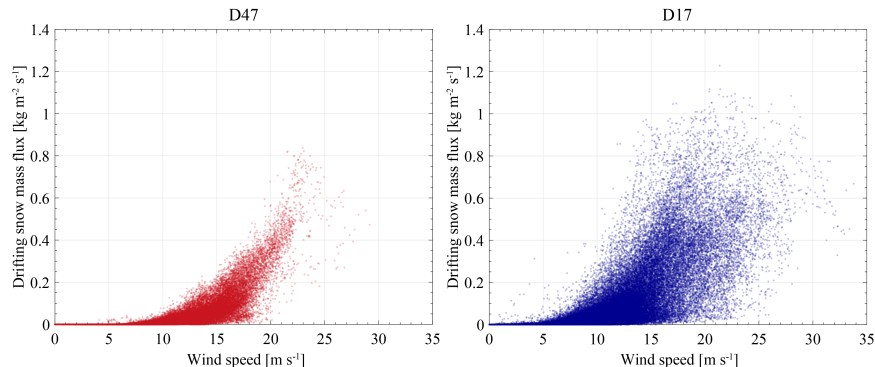

**Figure 5.** Drifting snow mass flux against 2-m wind speed recorded at D47 (left panel) and D17 (right panel). Only periods for which two 2G-FlowCapt™ sensors are installed and/or the lower sensor is not entirely covered with snow (i.e. $h_1 > 0.1$ m) are considered.

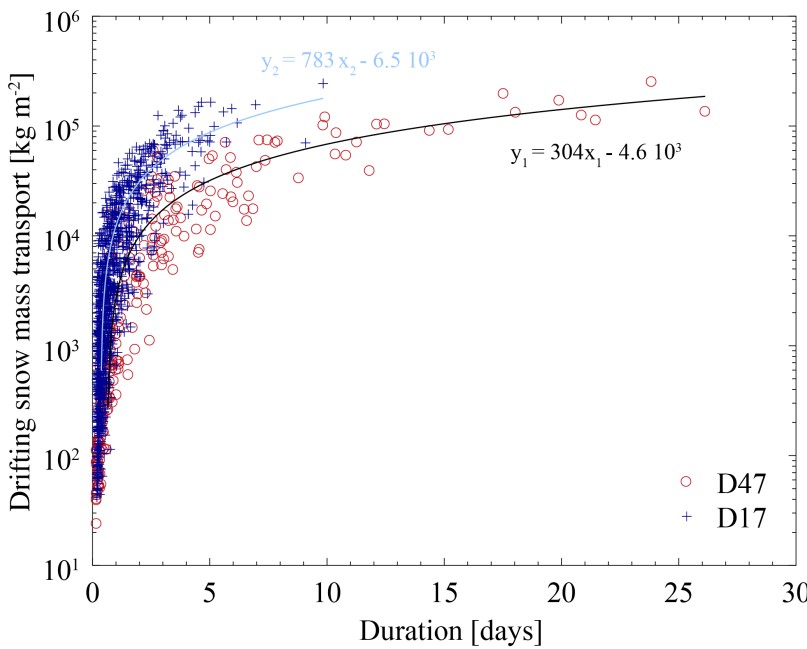

**Figure 6.** Logarithm of snow mass transport in drift conditions against duration for each drifting snow event recorded at D47 (red circles) and D17 (blue crosses). Only periods for which two 2G-FlowCapt™ sensors were installed and/or not entirely covered with snow are considered. Linear fits for D47 (black line) and D17 (light blue line) data are also reported on the graph, in which duration is expressed in hours.

is depicted (annual averages range from 0.73 to 0.85 at D47 and 0.45 to 0.68 at D17), drifting snow frequency varies strongly within the year, with an amplitude that can differ from year to year. Both locations experience a higher incidence of drifting snow in winter (defined here as the 8-month period between 1 March and 1 November) than during the rest of the year, a pattern
quite common over Antarctica (Mahesh et al., 2003; Scarchilli et al., 2010; Gossart et al., 2017; Palm et al., 2018). At the end of winter, a gradual decrease in drifting snow frequency is observed until a minimum is reached during summer, consistently with the annual course of wind speed (see Fig. S2, upper panel). This seasonal contrast is more pronounced at D17 than at D47 due to the stronger inhibition of erosion in summer resulting from lower wind speeds and higher air temperatures that promote the formation of cohesive bonds holding particles to the surface (e.g., Schmidt, 1980; Amory et al., 2017). Although
the use of a lowered threshold does not affect significantly the derived frequency, the stronger sensitivity to the increased threshold demonstrates the important contribution of occurrences of relatively small magnitude (i.e. $< 10^{-2}$ kg m$^{-2}$ s$^{-1}$ to the overall frequency. This demonstrates the need to specify explicitly the chosen threshold value when computing drifting snow frequency from 2G-FlowCapt™.

     Higher monthly values of drifting snow frequency are also systematically observed 100 km inland at D47 than close to the
coastline at D17. Analysis of drift conditions documented simultaneously at D17 and D47 for the 3-year period 2010-2012 shows a significant spatial variability, with almost all drifting snow occurrences at D17 involving drifting snow at D47 while

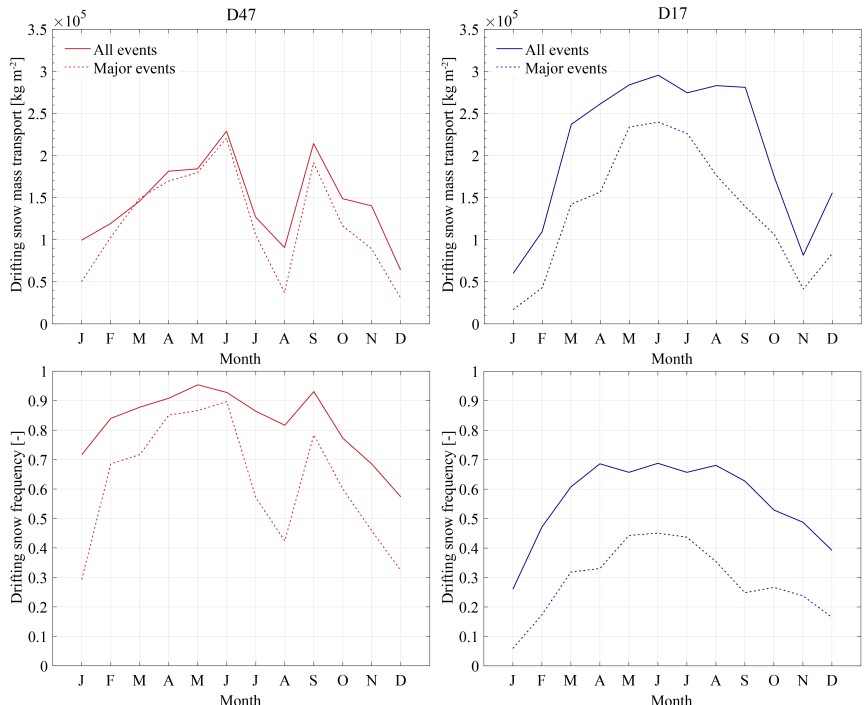

**Figure 7.** Intra-annual variability of drifting snow mass transport (upper panels) and related drifting snow frequency (lower panels) at D47 (left panels) and D17 (right panels). The relative contribution of major drifting snow events (see text) is highlighted in dotted lines. Summed mass transport and frequency values have been first determined for each month, and averaged within each monthly bin to produce monthly average values. The variability (standard deviation) in not shown due to the short length of the timeseries.

the opposite does not hold true (Table S2). Wind speeds at D47 for which drifting snow is observed at D47 only (28.3 % of occurrences) are generally lower (average of 11.5 m s$^{-1}$ compared to those for which the two locations experience drifting snow simultaneously (average of 13.5 m s$^{-1}$). This means that the largest occurrences are seen at both sites, and the higher drifting snow frequency at D47 is mainly due to additional occurrences of lesser magnitude for which the reduced wind speed downstream at D17 is not high enough to trigger snow transport.

### 3.2 Frequency of occurrence

Wind speeds at 2-m height for which drifting snow is detected (averages of 13.1 m s$^{-1}$ for D47 and 12.6 m s$^{-1}$ for D17) are generally higher than those occurring without drifting snow (averages of 7 m s$^{-1}$ for D47 and 6.1 m s$^{-1}$ for D17), although a wide range of similar wind speeds coexists between both categories. Following the approach of Baggaley and Hanesiak (2005) aiming at predicting the occurrence of snow transport from a set of common meteorological parameters, a credibility index (CRED) was used in a simpler approach to provide an estimation of the frequency of occurrence of drifting snow under specific

wind conditions

$$CRED = \frac{p}{p+n} \tag{3}$$

where p is the number of occurrences of drifting snow for a given wind speed range and n is the number of non-occurrences within that range. CRED varies from 0 to 1 and reflects the probability of observing drifting snow for a given range of wind speeds. Here a CRED of 0 means that no occurrence of drifting snow was observed for the selected range of wind speeds, while a CRED of 1 indicates that all wind speeds in that range were associated with drifting snow. CRED was calculated from the meteorological dataset within 1 m s$^{-1}$ wide intervals of wind speed. Occurrences observed below 2 m s$^{-1}$ and above 22 m s$^{-1}$ were not considered since their relative proportion within each wind speed interval individually accounted for less than 1 % of the observations. As in Fig. 2 the sensitivity of CREDs to the relaxed and stricter confidence thresholds used for acknowledging the occurrence of drifting snow is illustrated by the shaded areas.

The frequency of occurrence generally increases with wind speed (Fig.3) and typically resembles a cumulative normal distribution (Baggaley and Hanesiak, 2005). As the 2G-Flowcapt™ does not provide information on the source of windborne snow particles, CREDs in wind speed intervals lower than 5 m s$^{-1}$ at D17 most likely correspond to rare occurrences detected during snowfall (without necessarily involving erosion of snow) or shortly after the deposition of a loose snow layer easily erodible during light-wind conditions. Then, for higher intervals, small differences in wind speed involve large variations in the CRED. At both sites, the likelihood of observing drifting snow becomes important (CRED > 0.5) when wind speeds rise above 8 m s$^{-1}$. Wind speeds above 12 m s$^{-1}$ almost systematically produce drifting snow (CRED > 0.9), indicating that threshold (friction velocity) values for snow transport are most often exceeded in such wind conditions. Differences in local climate between the two locations could be expected to affect CRED distributions through their influence on post-depositional processes. Lower average wind speeds at D17 could be associated with lower compaction rates enabling drifting snow to be triggered by lower wind speeds than at D47. Conversely lower drift-induced compaction at D17 could be compensated by stronger interparticle bonding resulting from higher average air temperatures. Accurate investigation of these aspects would inexorably require knowledge of snow properties at the surface. However, Fig. 3 illustrates substantially similar CRED distributions, indicating that wind speed is the main driver behind the occurrence of drifting snow at these locations.

### 3.3 Duration of drifting snow events

Drifting snow occurs as long as the effective shear stress exerted on the snow surface by the overlying airstream, (i.e. the friction velocity) equals or exceeds the threshold value for erosion. Concurrent snowfall and advection of snow from upwind areas can also contribute to the windborne snow mass. The incidence of drifting snow depends on (and is affected by changes in) flow dynamics, surface roughness, cohesion of exposed surface snow particles or more generally availability of erodible snow, the combination of which determining the magnitude of snow mass fluxes and the duration of drifting snow events (Vionnet et al., 2013; Amory et al., 2016, 2017).

Following Vionnet et al. (2013), a drifting snow event has been defined as a period over which snow transport is detected for a minimum duration of 4 hours. That is, an event is considered to start and end when the half-hourly snow mass flux at

the lower unburied level $\eta_i$ respectively rises and drops below the confidence threshold of $10^{-3}$ kg m$^{-2}$ s$^{-1}$. To focus on significant drifting snow events, an additional criterion requires that a snow mass $Q_{DR}$ of at least 15 kg m$^{-2}$ (resulting from a drifting snow mass flux $\eta_{DR}$ at the confidence threshold of $10^{-3}$ kg m$^{-2}$ s$^{-1}$ for a duration of 4 hours) is transported along the event. Note that in case of complete burial of the lower sensor (i.e. $h_1 < 0.1$ m) or in the absence of sensor at the second level, this criterion is applied on the snow mass transport computed from the single available level without correction. By applying this selection procedure to the whole database, 1566 and 226 drifting snow events have been respectively identified at D17 and D47. Most events do not exceed 72 hours at D17 and can reach 10 days at most while a slight proportion (7 %) of events at D47 lasts more than 10 days with a maximum duration of 26 days (Fig. 4). In short, drifting snow events are on average twice as numerous but roughly two times shorter at D17 (yearly average number of 173 and median duration of 15 hours) than at D47 (yearly average number of 95 and median duration of 27.5 hours) where stronger winds can sustain longer events. Note that these statistics are not significantly altered if the length of the timeseries considered for D17 is reduced to that of D47.

### 3.4 Horizontal drifting snow mass transport

The drifting snow mass flux $\eta_{DR}$ typically tends to increase with wind speed in a power-law fashion (Fig. 5). This well-known behavior (Radok, 1977; Mann et al., 2000) is however depicted with significant dispersion and notable differences between the two locations; the data at D17 show that drifting snow mass fluxes can be of greater magnitude than at D47 for similar wind speeds and exhibit a generally higher variability along the range of wind speeds. This illustrates the diversity and spatial variability in factors controlling the windborne snow mass, as mentioned in the previous section. While wind speed can be used to predict the occurrence of drifting snow with a quite similar probability distribution between both locations (Fig. 3), on the other hand Fig. 5 demonstrates that more caution should be taken when scaling drifting snow mass transport with wind speed or related single parameter independent of surface snow properties (e.g., Mann et al., 2000). Such an approach would indeed involve mixtures of power laws to capture the large variability in drifting snow mass flux within the same wind speed interval, particularly at D17 where almost the entire range of values is observed from 15 m s$^{-1}$. Drifting snow is highly non-linear in nature and results essentially from the competitive balance between atmospheric drag and cohesive forces acting on the snow surface. This means that concurrent documentation of turbulence and surface snow properties are required for a better assessment of drifting snow processes and improvements of model predictability (e.g., Baggaley and Hanesiak, 2005; Vionnet et al., 2013).

Despite the non-linear behaviour of the drifting snow mass flux illustrated in Fig. 5, $Q_{DR}$ increases linearly with the event duration (Fig. 6). Values of $Q_{DR}$ have been computed for each drifting snow event identified in the database along which data from the two sensors are uninterruptedly available. Linear regression fits are shown and their respective equation are reported on the graph. A logarithm scale is preferred for readability purposes. Figure 6 also shows that $Q_{DR}$ hardly exceeds $10^5$ kg m$^{-2}$ even for the longest events, which thus seems to appear as an upper bound value for the mass transported in drift conditions during a single event. This is particularly well illustrated by D47 data. High values of $Q_{DR}$ for a wide range of durations involve large snow mass fluxes recorded at the two measurement levels, indicating the regular occurrence of well-developed, non-intermittent transport events in which particles are simultaneously carried out through both the saltation and suspension

mechanisms. This suggests that events of small magnitude involving low values of $Q_{DR}$ and/or during which transport in saltation dominates over transport in suspension must be comparatively short-lived. This however cannot be substantiated by studying the two levels separately because snow mass fluxes are vertically integrated over the exposed length and the sensor, which for the sensor closest to the ground almost always largely exceeds typical saltation heights (i.e. $\sim$0.1 m).

On an annual basis, both kinds of events combine to produce yearly values of $Q_{DR}$ close to or above 2 $10^6$ kg m$^{-2}$ at both locations (Table 2). Note that annual values of $Q_{DR}$ decrease only very slightly (less than 5 %) when a stricter confidence threshold is applied (i.e. only snow mass fluxes $\eta_i > 10^{-2}$ kg m$^{-2}$ s$^{-1}$ are considered), a result of large snow mass fluxes well beyond this threshold regularly occurring during drifting snow events. Such high estimates suggest that redistribution of snow by wind together with concurrent sublimation of snow particles during transport are important components of the surface mass balance in Adelie Land (Agosta et al., 2012; Amory and Kittel, 2019).

## 3.5   Contribution of major drifting snow events

The linear relationship between $Q_{DR}$ and event duration illustrated in Fig. 6 can be used to distinguish the contribution of the largest events to the drifting snow mass transport from that of the residual events. Major drifting snow events have been defined as the events whose duration is higher than the $75^{th}$ percentile for each site. Figure 7 shows that such major events, preferably but not exclusively grouped in winter, account for a reduced proportion of the overall events (resp. 22 % and 24 % for D47 and D17) but mainly dictate the variability of $Q_{DR}$ at the monthly scale, with the largest winter events capable of transporting alone up to 9 % of the annual quantity. The average monthly frequency resulting only from the occurrence of major events in each month is reported on the graph. As mentioned above, only the periods for which the snow mass flux was measured continuously at two levels have been considered. Note that this requirement is met for distinct periods of time between both measurement locations which thus must not be compared directly. At D17 (7), right panels), major events account for about half of the observed frequency but contribute to a larger part (> 70 %) of the mass transported in drifting snow. Larger monthly values of $Q_{DR}$ in winter result from an increased occurrence of major events combined with stronger snow mass fluxes (Amory et al., 2017), while drifting snow in summer mainly occurs in the form of residual events of lower magnitude. The data collected at D47 (Fig. 7, left panels) indicate that major events can contribute to an even larger part (> 82 %) of the annual transport and bring a different general perspective by showing that $Q_{DR}$ can be as important in summer than during some winter months, depending on the occurrence of major events. Despite a high and relatively uniform incidence of drifting snow in winter, the sharp decrease in $Q_{DR}$ from June to August at D47 is due to a reduced occurrence of major events during this period. This demonstrates that high monthly values of drifting snow frequency do not directly relate to the magnitude of snow transport since they can mainly consist of multiple but relatively brief events involving low or moderate snow mass fluxes. This also suggests that, in a modelling perspective, representing these major events rather than the complete range of drifting snow occurrences would be sufficient to capture the bulk of the contribution of drifting snow processes to the local surface mass balance.

## 4 Conclusion

Meteorological data and snow mass fluxes automatically acquired at two locations 100 km apart in Adelie Land, D17 and D47, have been combined to illustrate the spatial and temporal variability in drifting snow frequency and mass transport in a small portion of the East Antarctic coast. While the equipment at D47 has been dismantled after a period of 3 years (2010-2012), station D17 is still operative and the data provided nearly continuously for a period of 9 years (2010-2018) constitute the longest database of autonomous near-surface measurements of drifting snow currently available over the Antarctic continent. It should be noted that data collection continues at D17 and new measurements will be available in the future. Statistical analysis of the current dataset indicates that the likelihood of drifting snow increases with wind speed. Drifting snow occurred 81 % and 57 % of the time on average at D47 and D17, with maximum and minimum frequency values respectively observed in winter and summer in line with the annual course of wind speed. The higher drifting snow frequency at the more inland location D47 is most likely the result of locally higher wind speeds. Such high incidences of drifting snow and annual mass transport values reaching or exceeding $2 \; 10^6$ kg m$^{-2}$ at both sites suggest that drifting snow processes are important components of the local surface mass balance that would require a specific attention in a modelling context. By imposing a minimum duration of 4 hours and a minimum mass transport of 15 kg m$^{-2}$, 226 and 1566 drifting snow events have been detected at D47 and D17 over the respective observation periods. Events at D17 typically last 15 hours (median value) and are roughly twice shorter than at D47 where longer events can be sustained by higher wind speeds. The observations also demonstrate that most of the mass transported annually in drifting snow is carried out through a few major events accounting for less than 25 % of all the events and occurring preferably in winter, indicating that modelling the influence of drifting snow on the surface mass balance in this area might primarily rely on an accurate representation of these major events. The poor spatial and temporal coverage of satellite lidar techniques renders it difficult to determine the mean duration of snow-transport events such as those reported here. However, blowing snow events covering large areas can be successfully detected and tracked over a period of days as demonstrated in Palm et al. (2011). The presence of clouds impeding satellite retrieval is additionally responsible for the omission of overcast and/or snowfall conditions during which blowing snow is likely to occur preferentially because of the increased availability of loose snow. This can be particularly restrictive in coastal regions where the occurrence of blowing snow is often associated with synoptic-scale weather systems involving the presence of optically thick clouds (Gossart et al., 2017). The observations presented in this study, while providing spatially limited information, enable a continuous detection of snow-transport occurrences even in the presence of clouds and/or during snowfall. Although likely representative of local conditions, they constitute an original dataset dedicated to a poorly-documented, yet widespread feature of the Antarctic climate that can be used to complement satellite products and evaluate snow-transport models close to the surface and at high temporal frequency. Such exercises are needed to improve our understanding of the links between the occurrence and magnitude of drifting snow and ambient meteorological conditions, and ultimately better quantify the influence of drifting snow on the climate and surface mass balance of the Antarctic ice sheet.

*Data availability.* The database presented and described in this article (Amory et al., 2020) is available for download at https://zenodo.org/record/3630497

(last access: 29 January 2020). The data of the upcoming years will be added to the database on a yearly basis and made available to the community.

*Competing interests.* The author declares that he has no conflict of interests.

*Acknowledgements.* This work would not have been possible without the financial and logistical support of the French Polar Institute IPEV (program CALVA-1013). The author would like to thank all the on-site personnel in Dumont d'Urville and Cap Prud'homme for their

precious help in the field, in particular Philippe Dordhain for electronic and technical support. Christophe Genthon and Vincent Favier are also acknowledged for their investment in collecting data and maintaining the observation system in Adelie Land. The author thanks Christoph Kittel, Stephen Palm and two anonymous reviewers for providing many constructive comments. C. Amory is a Postdoctoral Researcher from the Fonds de la Recherche Scientifique de Belgique (F.R.S.-FNRS).

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
