# Peer review of "Drifting snow statistics from multiple-year autonomous measurements in Adelie Land, eastern Antarctica"

_The Cryosphere, 2019_

## Author Comment (AC1) · 6 Nov 2019

| Station | D17 | | | D47 | | |
|---|---|---|---|---|---|---|
| Variable | Period | Sensor | Accuracy | Period | Sensor | Accuracy |
| Wind speed | 2010
2011-2018 | RNRG 40C
A100LK[*] | $\pm 0.14$ m s$^{-1}$ at 10 m s$^{-1}$
$\pm 0.1$ m s$^{-1}$ | 2010-2012 | 05103-L R.M. Young | $\pm 0.3$ m s$^{-1}$ |
| Wind direction | 2010
2011-2018 | RNRG 200P
W200P[*] | $\pm 4$ °
$\pm 2$ ° | 2010-2012 | 05103-L R.M. Young | $\pm 3$ ° |
| Temperature | 2010-2018 | Vaisala HMP45 | $\pm 0.4$ °C at -20 °C | 2010-2012 | Vaisala HMP155 | $\pm 0.3$ °C at -20 °C |
| Relative humidity | 2010-2018 | Vaisala HMP45 | $\pm 2$ % (RH < 90 %) | 2010-2012 | Vaisala HMP155 | $\pm 1$ % (RH < 90 %) |
| Snow height | 2012-2018 | SR50[*] | $\pm 0.01$ m | 2010-2012 | SR50[*] | $\pm 0.01$ m |
| Drifting snow | 2010-2018 | 2G-FlowCapt™ | not specified | 2010-2012 | 2G-FlowCapt™ | not specified |

**Table S1.** Meteorological instruments installed at D17 and D47 along the respective observation periods (sensors marked with [*] are manufactured by Campbell Scientific, Inc.).

| D17
D47 | DS | nDS |
|---|---|---|
| DS | 54.8 % | 28.2 % |
| nDS | 2.8 % | 14.2 % |

**Table S2.** Comparison of drifting snow occurrences at D17 and D47 over the period 2010-2012.

[Figure]

**Figure S1.** Location of Dumont d'Urville station (DdU) and sites D17 and D47 in coastal Adelie Land and schematic cross-section showing elevation and distance from the coastline for each site. Coloured contours show elevation each 500 m from 0 to 4,500 m.

[Figure]

**Figure S2.** Pictures of the meteorological settings and wind roses at D17 (left panel) and D47 (right panel) for the respective observation periods. The wind direction at D17 is sampled at the upper level of the meteorological mast. The colours indicate the wind speed ranges in m s$^{-1}$.

[Figure]

**Figure S3.** Monthly timeseries of wind speed (upper panel), temperature (middle panel) and relative humidity (lower panel) at 2-m height for D47 (red circles) and D17 (blue squares) for the respective observation periods 2010-2012 and 2010-2018. Mean values for each variable have been first determined from the measurement level closest to 2 m for each month of the observation period, and averaged within each monthly bin to produce monthly average values. The variability (standard deviation) in not shown due to the short length of the timeseries.

[Figure]

**Figure S4.** Inter-annual (left panel) and intra-annual (right panel) variability of drifting snow frequency at D47 (red circles) and D17 (blue squares) for the respective observation periods 2010-2012 and 2010-2018. Mean values have been first determined for each month of the observation period, and averaged within each monthly bin to produce monthly average values. The variability (standard deviation) in not shown due to the short length of the timeseries.

---

## Referee Comment (RC1) · Anonymous Referee #1 · 19 Nov 2019

The manuscript describes a unique long dataset of blowing snow observations in Adelie Land, Antarctica, using weather stations that contained FlowCapt sensors for detecting blowing and drifting snow particles. Given that the surface mass balance in Antarctica is dominated by wind erosion and deposition (and thus blowing snow), in-situ measurements are very valuable. It's important that the manuscript is published, describing the setup, and providing some first analysis of frequency of blowing snow events, to increase the value of the dataset. The dataset may become useful for a broad community, including remote sensing and ice sheet (climate) modelling. I think the paper is well suited for TC. I have some minor comments which may help to improve the paper. A major concern I have is that the data is only available from the author on request,

whereas I think that the Copernicus Data Policy strongly discourages this. Using for example zenodo and doi versioning, the current dataset could be deposited there, and updated with newer data as a newer version of the data.

General comments

1) A better discussion of the accuracy of FlowCapt is required. In section 2.3, L140, the sensor is described as being accurate, and the only reference is Cierco et al. 2017). However, they write: "As a consequence, and even if the sensor provides good information in operational use, a regrettable inaccuracy in the collected data prevents the use of such measurements for research purposes. Nevertheless, a correction algorithm based on a statistical calibration of the sensor is proposed, which should make it possible to use the recorded data for preliminary approximations." In L151-153, it is discussed qualitatively, but can quantitative error margins be established? In any case, this section needs to be expanded, with a more detailed accuracy assessment of FlowCapt sensors.

2) It is not clear what the measurement heights were, particularly for D17. Also, it should be made clear throughout the manuscript which measurement level is used for wind speed, temperature and RH, at D17 when plotting or analyzing.

3) Fig. 2 shows that drifting snow occurs with lower wind speeds in D17 than in D47. Is this because the surface snow density, or bond strength is lower at this site? Lower average wind speeds can be associated with lower surface density. Or are the two measurement heights not comparable? It would make sense to also discuss the occurrence of drifting snow as a function of friction velocity, as mentioned in L238-239 which is often the variable of interest, determining whether or not snow erosion from the surface occurs. For D17, the profile measurements easily allow for a determination of friction velocity from the 6 measurement heights of wind speed. Similarly, L288, surface roughness could also be quantified from the wind speed profile, which could be used to demonstrate this relationship.

4) Eq. 2 confuses me, because the summation symbol lacks what it is summing over. At first, I thought that this was just how to compute half-hourly transported mass, but it seems to be for event transported mass and that the summation is over all the time steps constituting an event. In that case, L276 should introduce the definition of drifting snow event, if it is not each half-hourly data interval. Is the first half hourly interval that drops below the limit considered the end of the event? Nowhere in L262-271 is it introduced that there is a switch to the event based analysis.

5) Cierco et al. (2007) mentions that FlowCapt sensors can saturate. Could that be an explanation for the apparent upper bound on snow transport mass (L291-292)?

6) I'm not sure if the Online Supplement is necessary. It seems that the manuscript would benefit from inclusion of most of the materials in the main text. In particular, I think that the map with location of the stations should be part of the main text.

Minor comments:

L7: "hydrological"? Maybe "surface mass balance" is better (see L21-22)?

L8: "punctual" doesn't seem to be the right word here. I suggest: "model and satellite based products".

L13: "The data provided nearly continuously so far constitutes". Maybe add commas for clarification: "The data, provided nearly continuously so far, constitutes"

L25: It's confusing: wind confluence for me is the *convergence* of wind, unless there is compensating acceleration (which doesn't seem to be the case given that D47 has higher wind speeds than D17), so how does this relate to the horizontal *divergence* of snow?

L26-28: "Sublimation of snow particles ..." Please add citations. There are several good papers on drifting snow sublimation out there which deserve citation here.

L35: "are to be found" should that not be "are found"?

[Figure]

L36: "in the absolute values attributed to the relative contribution of these various mechanisms." Sounds vague.

L40: "3 times fewer" seems to refer to "mass fluxes". But either "3 times fewer" refers to occurrence, or it should be "3 times less".

L43: "The degree of plausibility of model-dependant features". I assume "model-dependent", but the sentence is a little vague.

L80-82: It reads as if the experiments were only in January 2010, and it should be highlighted here that the dataset is much longer. I would recommend writing something like: "For long term data acquisition following the measurement campaign, two distinct locations ..."

Section 2.1: How is the power supply organized? Can there by a bias in data availability depending on power source? (For example, if the battery tends to be drained towards the end of the winter season).

Section 2.1: It would be good to mention here explicitly that D17 is still operative.

Section 2.1: Fig. S2 in the supplement should show the dates on which the photos were taken.

L112: "the drainage of the sinking near-surface air" sounds vague to me.

L113: "over an unobstructed"

L118: "higher incidence of drifting snow", maybe add a reference to Fig. S4?

L119: "combined with"

L118-119: I suggest to reference the accompanying Brief Communication here. "Brief communication: Rare ambient saturation during drifting snow occurrences in coastal East Antarctica"

L160-161: This seems like an appropriate location to introduce Equation 2, instead

of Section 3.4. At least, I assume that the same correction is made when part of the FlowCapt was buried as done in Equation 2?

L171: Again, I don't think punctual is appropriate here.

L194-195: This statement deserves citations.

Fig. 1, as well as Fig. 2: The figure caption should also mention what the gray shaded areas denote (currently it's only explained in the main text).

L268: Maybe add: "h_ref = 2 m, which is the sum of two 1 m long FlowCapt sensors."

L290: This is confusing. Fig 4, right panel should be a non-log (i.e., linear) scale in order for it to show a linear increase of QT and event duration.

L296-297: Could this be substantiated by showing the two levels separately?

---

## Referee Comment (RC2) · Anonymous Referee #2 · 24 Nov 2019

This paper presents the analysis of 8 yr field observation of drifting snow at two sites D17 and D47 on Terre Adelie Land (east Antarctica). The main tools used in this study are FlowCapt acoustic sensor and associated Automatic Weather Station. The paper contributes to knowledge concerning the measurement of negative term of surface mass balance driven by wind.

The manuscript subject is appropriate for Cryosphere Journal, well written, data and analysis are very important. The data are partially already presented and analysed in previous paper (Trouvilliez et al, 2014 and 2015) and a paper under review on the Cryosphere (Amory & Kittel, submitted) presents the same data under the aspect of the

sublimation that is the main issue not discussed in this manuscript. The interpretations of data acquired are supported by the result and the amount of the good data, but the statistic analysis present in the manuscript are not relevant to support the publication on the high quality Journal such as "The Cryosphere" and the Surface Mass Balance condition from the previous studies are not taken adequately in account.

The previous paper on SMB survey at the two sites (D17 and D47) and model on the Adelie Coast are not adequately discussed and reported (example: Pettrè et al., 1986; Bintanja, 1998; Pourchet et al., 1997; Frezzotti el ta., 2004, Genthon et al., 2007; Agosta et al., 2011, Favier et al., 2013, Barral et al., 2014 in the reference but is not taken in account in the manuscript; Goursaud et al., 2017). The manuscript does not analysed the Surface Mass Balance and in particular the extensive presence of the blue ice area in the Coastal Terra Adélie (Favier et al., 2011) and their implication on the drifting snow.

The author does not distinguish drift from blowing snow phenomena and the threshold of snow sublimation, and their implication on the mass transport/sublimation and the difference between the two sites. Blowing and drifting snow are not redistribution process, a significant part of blowing snow sublimate as pointed out by snow radar survey (see Frezzotti et al., 2007; Eisen et al., 2008) or satellite survey (Scarchilli et al., 2010, Palm et al, 2011, 2017; Scambos et al., 2012).

The AWS and FlowCapt sensor provide single measurement point for limited number of years and must be analysed in the contest of Surface Mass Balance study derived from other field measurements as stakes, firn cores, snow radar profile and satellite studies.

---

## Author Comment (AC2) · 29 Jan 2020

I sincerely thank the reviewer for its thorough reading of the paper, the very relevant questions and the interesting suggestions that will undoubtedly help to improve the paper. My responses are reported hereafter in red.

**Response to reviewer RC1**

The manuscript describes a unique long dataset of blowing snow observations in Adelie Land, Antarctica, using weather stations that contained FlowCapt sensors for detecting blowing and drifting snow particles. Given that the surface mass balance in Antarctica is dominated by wind erosion and deposition (and thus blowing snow), in-situ measurements are very valuable. It's important that the manuscript is published, describing the setup, and providing some first analysis of frequency of blowing snow events, to increase the value of the dataset. The dataset may become useful for a broad community, including remote sensing and ice sheet (climate) modelling. I think the paper is well suited for TC. I have some minor comments which may help to improve the paper. A major concern I have is that the data is only available from the author on request, whereas I think that the Copernicus Data Policy strongly discourages this. Using for example zenodo and doi versioning, the current dataset could be deposited there, and updated with newer data as a newer version of the data.

The data has been deposited on a public repository via zenodo and can now be downloaded at https://zenodo.org/record/3630497. The section data availability has been modified accordingly.

Please note that the SPC data used for the evaluation of FlowCapt sensors as proposed in the revised version of the manuscript are currently involved in an ongoing publication and have thus not been deposited together with the drifting snow dataset on zenodo.

**General comments**

1) A better discussion of the accuracy of FlowCapt is required. In section 2.3, L140, the sensor is described as being accurate, and the only reference is Cierco et al. 2017). However, they write: "As a consequence, and even if the sensor provides good information in operational use, a regrettable inaccuracy in the collected data prevents the use of such measurements for research purposes. Nevertheless, a correction algorithm based on a statistical calibration of the sensor is proposed, which should make it possible to use the recorded data for preliminary approximations." In L151-153, it is discussed qualitatively, but can quantitative error margins be established? In any case, this section needs to be expanded, with a more detailed accuracy assessment of FlowCapt sensors.

All the work done by Cierco et al. (2007) on evaluating the reliability of the Flowcapt$^{TM}$ sensor and characterizing its limitations has focused on the first-generation device. The sensors installed at D17 and D47 are of a more recent design (referred to as second-generation FlowCapt or 2G-FlowCapt in the paper) which significantly improves, without necessarily solving, inaccuracy issues with estimating snow mass fluxes (Trouvilliez et al., 2015).

I have reorganized Section 2.3 to better structure the information provided on 2G-FlowCapt$^{TM}$ sensors and utilization of the data. The section is now divided in four parts whose one of them is dedicated to accuracy assessment of the sensors. More specifically I mention results (which I suggest to provide in supplementary materials with further details on the experimental set-up and methodology) from a intercomparison experiment I've led myself in Adelie Land between the 2G-FlowCapt$^{TM}$ and a snow particle counter (SPC-S7), an optical device considered as a kind of reference sensor for estimating snow mass fluxes (Sato et al., 1993). The field experiment took place at site D17 in late January 2014 and focused on a 24 hour long snow transport event. Although more data are necessary to better assess the performance of the 2G-FlowCapt$^{TM}$ in Antarctic conditions, the comparison shows a reasonable agreement between the two types of sensors (Fig. R1; proposed as Fig. S4 in supplementary materials). Reported below is the content of Sect. 2.3 in the revised version of the manuscript entitled "Accuracy assessment" and the related supplementary materials; please refer to the track changes for a complete report of the modifications undertaken in this section:

*"[…]*.

**2.3.3. Accuracy assessment**

*While FlowCapts™ sensors can detect the occurrence of snow transport with a high level of confidence, the ability of the original design to estimate snow mass fluxes is more questionable (Cierco et al., 2007). These accuracy issues, without being necessarily solved, have been significantly improved with the 2G-FlowCapt™, facilitating its use for quantitative applications (Trouvilliez et al. 2015). Although measurement uncertainty is not known, the 2G-FlowCapt™ was shown to generally underestimates the snow mass flux relatively to integrated estimates computed from optical measurements made with a snow particle counter S7 (SPC-S7; taken as a reference in the study) during a winter season in the French Alps, particularly during concurrent precipitation (Trouvilliez et al. 2015). During mixed drifting snow events when erosion occurs simultaneously with snowfall, the density of precipitating particles which have not reached the ground yet is lower than eroded, more rounded snow particles originating from the ground which have lost their original crystal shape and size through collision, sublimation and the thermal processes of metamorphism. For a given snow mass flux, the particles' momentum, and by extension the measured acoustic pressure, is therefore lower during a mixed drifting snow event than during an event predominantly driven by the erosion process. This results in an underestimation of the snow mass flux measured by the 2G-FlowCapt™during mixed events, with a magnitude depending on the relative proportion of eroded particles against fresh snow particles.*

*Environmental conditions influence greatly the estimation of the snow mass flux by the 2G-FlowCapt™. The intercomparison experiment in the Alps was done within a range of mass flux values (< 2.5 $10^{-2}$ kg $m^{-2}$ $s^{-1}$) significantly lower than those encountered in Adelie Land (see Sect. 3.4). In addition, comparatively stronger surface winds and lower temperature on the Antarctic ice sheet favor the breaking and rounding of snow particles. This suggests that the performance of the 2G-FlowCapt™ remains to be assessed in the extreme Antarctic environment, in which large proportions of small, rounded particles can be expected in drift conditions (i.e. within 2 m above ground) even with concurrent precipitating snow (Nishimura and Nemoto, 2005).*

*A field experiment involving measurements with SPC-S7 and 2G-FlowCapt$^{TM}$ sensors performed during a 24 hour long snow transport event was undertaken at site D17 in late January 2014 (Fig. S3). Strong drift conditions were observed with 2-m wind speeds and snow mass fluxes reaching up to 19 m $s^{-1}$ and 4 $10^{-1}$ kg $m^{-2}$ $s^{-1}$ respectively. Although the statistical representativeness of the results may be small due to the low amount of data collected during only one event, the comparison shows that the snow mass fluxes provided by the two types of sensors are very similar in magnitude (Fig. S4). Further details on the experimental set-up and comparison methodology are provided in supplementary materials (Sect. S1).*

[Figure]

**Figure R1**. Comparison between snow mass fluxes provided by 2G-Flowcapt™ sensors and computed from measurements made with snow particle counters (SPC-S7) during a snow transport event at site D17 in January 2014. A distinction is made between snow mass fluxes integrated over 0.1 to 1.1 m and 1.2 to 2.2 m above ground.

Supplementary materials:

**S1. Intercomparison between snow particle counters S7 and second-generation FlowCapt™ sensors during a drifting snow event in Adelie Land**

**1.1. Snow particle counters**

The measurement principle of the snow particle counter S7 (SPC-S7) follows an optical method based on the strong absorption of the infrared light by the snow. The diameter and number flux of snow particles are detected by their shadows on a super-luminescent diode sensor. Electric pulse signals corresponding to a snow particle passing through a sampling area of 50 mm² (2 mm in height and 25 mm in width) and whose voltage is directly proportional to the size of the particle are classified into 32 size bins from ~ 40 to 500 μm (Sato et al., 1993). This means that snow particles smaller than 40 μm remain undetected and snow particles larger than 500 μm are assigned to the maximum diameter class. Thanks to a self-steering vane the SPC-S7 measures perpendicularly to the horizontal wind vector the distribution size spectrum of snow particles every 1 s, from which the horizontal snow mass flux, η, can be computed assuming fully spherical snow particles with a density equal to that of ice as follows:

$$\eta = \sum_{i_d=1}^{32} \eta_d = \sum_{i_d=1}^{32} n_d \frac{4}{3}\pi \left(\frac{d}{2}\right)^3 \rho_i$$

with $\eta_d$ (kg m$^{-2}$ s$^{-1}$) the horizontal snow mass flux for the class of diameter d (m), $i_d$ the index  and $n_d$ the measured number flux of snow particles (part. m$^{-2}$ s$^{-1}$) for each of the 32 diameter classes, and $\rho_i$ the particles density (917 kg m$^{-3}$).

**1.2. Experimental set-up**

Two SPCs were installed on 28 January 2014 (Fig. S3) a few hours before strong drifting snow occurred in conjunction with strong katabatic winds reinforced by the passage of a low-pressure system off the Adelie Coast. The equipment was removed on 29 January once drifting snow ceased. One SPC was installed at a fixed position 1 m above the ground, while the position of the other was alternatively switched manually between 0.5 et 2 m above the ground every 1-2 hours. This was done in order to study the vertical gradient of the mass flux for two ranges of height (0.1-1.1 m and 1.2-2.2 m) above the snow surface for which 2G-FlowCapt™ measurements are also available for comparison. The high energy requirements of the SPCs (~ 15 W) were fulfilled by an electric generator that was housed together with the acquisition system in a mobile shelter downwind of the measurement structure. Only a few data are missing due to problem with the acquisition system of the SPC at the beginning of the experiment, resulting in an timeseries almost continuous along the event.

**1.3. Computation of integrated snow mass fluxes from SPC data**

According to the diffusion theory of drifting snow (Radok, 1977), the averaged drifting snow particle density (kg m$^{-3}$) in the diffusion layer can be approximated by a function of height. When the wind profile follows a power law, an expression for the vertical distribution of the snow mass flux $\eta(z)$ (kg m$^{-2}$ s$^{-1}$) writes

$$\eta(z) = az^{-b}$$

where a is the calibration parameter and b the exponent independent of height. These parameters were derived by regression from the data measured by the two SPCs (Trouvilliez et al. 2015), alternatively available for the two height ranges. Then, the half-hourly average of the horizontal snow mass flux vertically integrated over the corresponding height covered by the 2G-FlowCapt™ can be estimated. Because (i) snow depth measurements revealed insignificant height change after the event and were affected by the presence of drifting snow particles perturbing the travel of ultrasound pulses along the measuring path during the event, (ii) the two 2G-Flowcapts were respectively installed at 0.1 and 1.2 m above the snow surface at the beginning of the event, and (iii) the heights of the SPCs were regularly checked and manually adjusted along the experiment, constant heights are used in the integration. Finally, data were processed following the procedure described in Guyomarc'h et al. (2019).

Resulting integrated snow mass fluxes are compared in Fig. S4. Although more data are necessary to better assess the performance of the 2G-FlowCapt$^{TM}$ in Antarctic conditions, a high degree of agreement between the two types of sensor is depicted with a correlation coefficient of 0.82 and 0.93 and a rmse of 70 10$^{-2}$ and 13 10$^{-2}$ kg m$^{-2}$ s$^{-1}$ (by taking the SPC-S7 as a reference) for the lower and upper measurement range, respectively.

[Figure]

*Figure S3. Picture of the snow particle counters installed at D17 during the intercomparison experiment in late January 2014.*

*Figure S4. See Fig. R1.*

2) It is not clear what the measurement heights were, particularly for D17. Also, it should be made clear throughout the manuscript which measurement level is used for wind speed, temperature and RH, at D17 when plotting or analyzing.

At D17 I've selected the measurement height closest to 2-m as recovered by the snow depth ranger or using the closest original height when no information on snow height is available. This is now explicitly detailed in the manuscript and specified in required places.

3) Fig. 2 shows that drifting snow occurs with lower wind speeds in D17 than in D47. Is this because the surface snow density, or bond strength is lower at this site? Lower average wind speeds can be associated with lower surface density. Or are the two measurement heights not comparable? It would make sense to also discuss the occurrence of drifting snow as a function of friction velocity, as mentioned in L238-239 which is often the variable of interest, determining whether or not snow erosion from the surface occurs. For D17, the profile measurements easily allow for a determination of friction velocity from the 6 measurement heights of wind speed. Similarly, L288, surface roughness could also be quantified from the wind speed profile, which could be used to demonstrate this relationship.

The definition of a drifting snow occurrence in the former version of the manuscript involved a drifting snow mass flux above the confidence threshold at either measurement level. After some reconsideration motivated by your comment on the lack of clarity about the definition of a drifting snow event, I figured out that this may still generate few artificial occurrences when fluxes at the upper level are above the threshold value while fluxes at the lower one are not for small values oscillating around the threshold value. I have then slightly modified the definition of a drifting snow occurrence and redone the analysis with the following definition:

"[…] *drifting snow has been considered to occur when the half-hourly mean of the snow mass flux exceeds a confidence threshold of $10^{-3}$ kg m$^{-2}$ s$^{-1}$ as determined from visual observations on the field in Adelie Land (Amory et al., 2017). Note that the same confidence threshold yielded a high level of agreement (98.6 %) between the SPC-S7 and 2G-FlowCapt$^{TM}$ in terms of occurrence detection in the comparison study led by Trouvilliez et al. (2015) in the Alps. Since this value remains small compared to snow mass fluxes estimated during drifting snow occurrences (see Sect. 3.4), the confidence threshold is assumed independent on the exposed length of the sensor.*

*The sensor is considered unburied as long as at least 10 % (i.e. 0.1 m) of its initial length remain uncovered with snow.*"

The resulting CRED distribution is presented in Fig. R2. Note that this figure is proposed in replacement of Fig. 2 in the revised version of the manuscript.

Indeed lower surface densities that can be expected at D17 due to lower wind speeds could enable drifting snow to occur at lower wind speeds than at D47. Conversely lower drift-induced compaction at D17 could be compensated by stronger interparticle bonding resulting from higher average temperatures. There is unfortunately no direct way to investigate that aspect, and answering your question in a more exhaustive way would inexorably require knowledge of snow properties at the surface, or at least determination of threshold friction velocities at D47, which are either not available or possible due to the single measurement level at D47. I then suggest to leave this question open for further studies. Anyways, updated Fig. R2 no longer suggests that such processes may be responsible for significant differences in CRED distribution between the two locations despite the actual differences in climate conditions, and indicates that wind speed is the main deriver behind the occurrence of drifting snow.

Accurate determination of friction velocity values from the data collected in the extreme environment of D17 involve several selection criteria (see Amory et al. 2017) that are far from being systematically met along the measurement period, resulting in discontinuous time series that can be additionally biased by an over-representation of climate conditions for which these criteria are met but not necessarily representative of the average climate conditions. In particular, the validity of Monin-Obukhov similarity theory, from which friction velocity and roughness length can be determined using the wind speed profiles, is questionable during drifting snow conditions. Another influent factor in the determination of friction velocity that could impede the generation of homogeneous time series is the number of anemometers available for each wind profile, which varies non-uniformly along the measurement period depending on instrument failure and burial of the lower levels. Note that evolution of the measurement height is also not known before 2013 at site D17. These factors, together with additional parameters such as the choice of the stability correction function, all justify a sensitivity study that lies beyond the scope of the present paper. A non-negligible issue when using the wind profiles at D17 to compute friction velocity values is the artificial roughness created during summer visits and maintenance/replacement operations. This is especially true since summer 2014-2015 when the meteorological mast at D17 was mounted on a sledge that was further buried under the snow. The whole operation, ideally repeated each year, requires excavation of several meters in depth within the snowpack and use of snow trucks that disturb surface roughness in the immediate vicinity of site D17 for an unknown duration.

For all these reasons, and also because site D47 is equipped with only one measurement level thus precluding spatial comparison with D17, I chose to focus on wind speed rather than friction velocity, enabling the discussion of continuous, more homogeneous times series at both locations simultaneously.

Similarly, threshold friction velocity would need to be continuously known all along each drifting snow event to investigate differences with friction velocity, so the theoretical statement made in L288 cannot be illustrated with the present data. It has therefore been removed from the text (see my response to the next comment). Here in the absence of direct measurements of surface snow properties, threshold friction velocity values can only be determined at the onset and end of drifting snow event, when the snow mass flux rises or drops below the confidence threshold value and only if friction velocity can be accurately determined from the wind speed profiles at the same time.

[Figure]

**Figure R2.** CRED distribution for drifting snow occurrences showing the increasing probability of observing drifting snow with increasing 2-m wind speed at sites D47 (red curve) and D17 (blue curve). Shaded areas correspond to CREDs respectively computed using a relaxed and a stricter confidence threshold of $10^{-4}$ kg m$^{-2}$ s$^{-1}$ and $10^{-2}$ kg m$^{-2}$ s$^{-1}$ and are shown as a measure of uncertainty.

4) Eq. 2 confuses me, because the summation symbol lacks what it is summing over. At first, I thought that this was just how to compute half-hourly transported mass, but it seems to be for event transported mass and that the summation is over all the time steps constituting an event. In that case, L276 should introduce the definition of drifting snow event, if it is not each half-hourly data interval. Is the first half hourly interval that drops below the limit considered the end of the event? Nowhere in L262-271 is it introduced that there is a switch to the event based analysis.

Several modifications have been made in order to improve clarity when dealing with the computation of snow mass fluxes and snow mass transport and are reported below:

 (i) Derivation of snow mass transport from former Eq. 2 has been re-expressed in a clearer way following two equations, starting from the formulation of the drifting snow mass flux $\eta_{DR}$ (i.e. vertically integrated between 0 and 2 m over the snow surface) as

$$\eta_{DR} = \begin{cases} \eta_1 + \eta_2, & h_1 + h_2 \geq h_{ref} \\ \eta_1 + \eta_2 \cdot \dfrac{h_{ref}}{h_1 + h_2}, & h_1 + h_2 < h_{ref} \end{cases} \tag{1}$$

*where $\eta_i$ (kg m$^{-2}$ s$^{-1}$) is the observed snow mass flux integrated over the exposed height $h_i$ (m) of the corresponding 2G-FlowCapt™ sensor, and $h_{ref} = 2$ m is the sum of two fully exposed, 1 m long 2G-FlowCapt™ sensors. In other words, when $h_1 + h_2 < 2$ m, it is assumed that the measured snow mass flux is constant up to 2 m. To keep consistency with the confidence threshold for the detection of drifting snow occurrences, snow mass fluxes below $10^{-3}$ kg m$^{-2}$ s$^{-1}$ have been set to 0. The horizontal drifting snow mass transport for a given period of time $[t_0, t_n]$, $Q_{DR}$, then writes*

$$Q_{DR}(t) = \int_{t_0}^{t_n} \eta_{DR}(t)dt. \tag{2}$$

 (ii) Following your recommendation, Eqs. 1 and 2 are now introduced in the methods section,

(iii) The definition of a drifting snow event in Sect. 3 has been made clearer and is now presented as "*a period over which snow transport is detected for a minimum duration of 4 hours. That is, an event is considered to start and end when the half-hourly snow mass flux at the lower unburied level $\eta_i$ respectively rises and drops below the confidence threshold of $10^{-3}$ kg m$^{-2}$ s$^{-1}$.*",

(iv) Complementary information has been given on the threshold used for acknowledging the occurrence of drifting snow in Sect. 2.3 as detailed in my response to general comment #3,

(v) All along the manuscript it is now explicitly mentioned to which snow mass flux (raw sensor output $\eta_i$ or corrected drifting snow mass fluxes $\eta_{DR}$) I refer to.

The relation between snow mass transport and wind speed is now discussed by studying half-hourly drifting snow mass fluxes computed from Eq. (1) as a function of related 2-m wind speeds (Fig. R3 – Fig. 5 in the revised version of the manuscript) instead of mass transport and average wind speed per event, and consists in the following analysis in the paper:

"*The drifting snow mass flux $\eta_{DR}$ typically tends to increase with wind speed in a power-law fashion (Fig. 5). This well-known behavior (Radok, 1977; Mann et al., 2000) is however depicted with significant dispersion and notable differences between the two locations; the data at D17 show that drifting snow mass fluxes can be of greater magnitude than at D47 for similar wind speeds and exhibit a generally higher variability along the range of wind speeds. This illustrates the diversity and spatial variability in factors controlling the windborne snow mass, as mentioned in the previous section. While wind speed can be used to predict the occurrence of drifting snow with a quite similar probability distribution between both locations (Fig. 3), on the other hand Fig. 5 demonstrates that more caution should be taken when scaling drifting snow mass transport with wind speed or related single parameter independent of surface snow properties (e.g. Mann et al., 2000). Such an approach would indeed involve mixtures of power laws to capture the large variability in drifting snow mass flux within the same wind speed interval, particularly at D17 where almost the entire range of values is observed from 15 m s$^{-1}$. Drifting snow is highly non-linear in nature and results essentially from the competitive balance between atmospheric drag and cohesive forces acting on the snow surface. This means that concurrent documentation of turbulence and surface snow properties are required for a better assessment of drifting snow processes and improvements of model predictability (e.g. Baggaley and Hanesiak, 2005; Vionnet et al., 2013).*".

[Figure]

**Figure R3**. Drifting snow mass flux against 2-m wind speed recorded at D47 (left panel) and D17 (right panel). Only periods for which two 2G-FlowCapt™ sensors were installed and/or the lower sensor was not entirely covered with snow (i.e. $h_1 > 0.1$ m) are considered.

As the definition of a drifting snow event is explicitly given in Section. 3.3, make a recall just a few paragraphs below would thus sound redundant. The paragraph describing the relation between mass transport and event duration introduces the switch to the event-based the analysis in the 2$^{nd}$ sentence: "*Values of $Q_{DR}$ have been computed for each drifting snow event identified in the database [...]*".

5) Cierco et al. (2007) mentions that FlowCapt sensors can saturate. Could that be an explanation for the apparent upper bound on snow transport mass (L291-292)?

If saturation of the sensors would be responsible for the apparent upper bound in snow mass transport, it should be apparent from the raw sensor outputs as well. However no evidence of such a behavior is found when plotting the half-hourly snow mass fluxes against wind speed as proposed in the new analysis (see the previous comment - Fig. R3). Note also that the evaluation of Cierco et al. (2007) makes use of the original design of the FlowCapt sensor, and the analysis proposed here relies on a more recent design which significantly improves, although not necessarily solve, inaccuracy issues with estimating snow mass fluxes (Trouvilliez et al., 2015).

6) I'm not sure if the Online Supplement is necessary. It seems that the manuscript would benefit from inclusion of most of the materials in the main text. In particular, I think that the map with location of the stations should be part of the main text.
Yes I agree, the location of the measurement sites is of first importance. The map has been included in the main text. I'd like to stress that I have a limited budget for this publication and inclusion of figures in the main text rather than in supplement goes with significant charges. So I'd like to stick to the very essential material in the main text if it seems reasonable, and leave the rest (Tables S1,S2 and Figs. S2,S2) as supplementary materials since they contain additional information that would surely be beneficial but are not really essential to the analysis.

**Minor comments**
L7: "hydrological"? Maybe "surface mass balance" is better (see L21-22)?
Thanks for the suggestion. Surface mass balance is the resultant of the hydrological cycle at the surface of the ice sheet, so I'd rather keep this formulation as it is, if it does not result in any misunderstanding.

L8: "punctual" doesn't seem to be the right word here. I suggest: "model and satellite based products".
I simply removed "punctual" and now speak about "model and satellite products".

L13: "The data provided nearly continuously so far constitutes". Maybe add commas for clarification: "The data, provided nearly continuously so far, constitutes"
Corrected accordingly.

L25: It's confusing: wind confluence for me is the *convergence* of wind, unless there is compensating acceleration (which doesn't seem to be the case given that D47 has higher wind speeds than D17), so how does this relate to the horizontal *divergence* of snow?
I agree that these two terms might be confusing when employed together as they may apparently relate to opposite situations. In a general picture, convergence of the wind field is associated with deceleration and thus deposition of snow (or convergence of snow by wind transport). While this is certainly verified over a flat surface, this does not however preclude the occurrence of erosion within steep regions of the confluence area, where katabatic winds converge from a large-scale perspective and still can accelerate locally depending on the slope of the ice surface.
Changes in surface slope play a crucial role in controlling locally the erosion/deposition process through acceleration/deceleration of the near-surface flow. This is illustrated through sub-kilometer spatial heterogeneities in accumulation in, for instance, coastal Adelie Land (Agosta et al., 2012), or alternating ridges and glazed surfaces scattered over East Antarctica (Scambos et al., 2012).
In an even more general picture, erosion occurs at every places where $u_*$ exceeds $u_{*t}$, whether the near-surface wind field displays a convergent or a divergent character. Wind confluence zones result from the large-scale interactions between the topography and near-surface flow. Cold-air drainage currents converge from a large interior area in smaller areas that thus drain the air from a much wider upstream reservoir. Katabatic confluence areas such as Adélie Land are generally characterized by stronger and more persistent winds (Bromwich and Liu, 1996; Parish and Bromwich, 2007), which can thus lead to enhanced snow transport in these regions, erosion where $u_* > u_{*t}$, or in other words, horizontal divergence of the wind transport of snow.
Nevertheless, for clarity I have removed the terms "divergence" and "convergence" from the paragraph you were confused by, and rewritten the sentence as "*In coastal areas, wind redistribution of snow is responsible for an export of mass beyond the ice-sheet margins.*". Note however that Adelie Land is still introduced as a wind confluence area of East Antarctica in the last paragraph of the introduction.

L26-28: "Sublimation of snow particles ..." Please add citations. There are several good papers on drifting snow sublimation out there which deserve citation here.

I originally intended to have this assertion supported by the several papers I refer to at the beginning of the next paragraph but yes, you're entirely right. I have added 3 significant studies (Mann et al., 2000; Bintanja, 2001; Thiery et al., 2012) highlighting and discussing the role of windborne snow sublimation in Antarctica.

L35: "are to be found" should that not be "are found"?

See our response to the next comment.

L36: "in the absolute values attributed to the relative contribution of these various mechanisms." Sounds vague.

I have reformulated the sentence as "Conversely, contrasting results can be found from one study to another in the absolute values attributed to the contribution of wind-driven processes to the large-scale mass transport."

L40: "3 times fewer" seems to refer to "mass fluxes". But either "3 times fewer" refers to occurrence, or it should be "3 times less".

"3 times fewer" does actually refer to "mass fluxes". I have reversed the sentence and written "3 times larger".

L43: "The degree of plausibility of model-dependant features". I assume "model-dependent", but the sentence is a little vague.

I have reformulated the sentence as "model results as well as the assumptions made in the implementation of wind-driven snow physics need to be carefully assessed with independent observations."

L80-82: It reads as if the experiments were only in January 2010, and it should be highlighted here that the dataset is much longer. I would recommend writing something like: "For long term data acquisition following the measurement campaign, two distinct locations ..."

I have changed the two following sentences to account for your suggestions: *"[…] a field campaign specifically dedicated to drifting snow has been initiated […]. Two distinct locations, namely D17 and D47 (Fig. S1), were instrumented for long term-data acquisition and equipped with […]."*

Section 2.1: How is the power supply organized? Can there by a bias in data availability depending on power source? (For example, if the battery tends to be drained towards the end of the winter season).

A power consumption balance has been calculated before installation of each stations according to the whole set of instruments to ensure a continuous power supply throughout the year. Each installation of new instruments (such as for instance, a second 2G-FlowCapt™ sensor and a sonic depth ranger at D17 in late December 2012) came along with installation of new sets of batteries and solar panels to supply for the additional power requirements. The batteries are also checked during each summer visit and replaced when needed. The batteries' voltage is continuously monitored and stored together with the meteorological variables in the datalogger to ensure that the entire system has been designed properly and is sufficiently supplied with energy throughout the winter. Note also that the FlowCapt™ sensors are known to be low-consuming and are moreover continuously solicited by the datalogger (RS232 connection), so we can easily distinguish between instrument failure (absence of response – no data) and data containing null values (absence of drifting snow). Except for the months of May and June at site D47 and a few other instances of malfunction scattered outside of maintenance operations, the dataset is continuous along the measurement periods. I have added the following related lines in the text:

*"The FlowCapt™ is low-power consuming and designed to withstand harsh climate conditions without regular human attendance. At each station battery voltage is monitored and stored together with the meteorological variables in the datalogger to ensure that the entire measurement system is sufficiently supplied with energy throughout the winter. The 2G-FlowCapt™ are continuously solicited by the datalogger (RS232 connection), such that instances of instrument malfunction (absence of response and no data) can be*

*unambiguously distinguished from the absence of drifting snow (data containing null values). A thorough check on the observations was performed and resulted in omission of misleading data wherever necessary. Except for those very few cases, maintenance periods in summer and a major 2-month failure of the lower 2G-FlowCapt™ sensor at D47 in May and June 2012, the dataset is continuous along the respective measurement periods.".*

Section 2.1: It would be good to mention here explicitly that D17 is still operative.
It is now explicitly mentioned in the section and in Table 1.

Section 2.1: Fig. S2 in the supplement should show the dates on which the photos were taken.
Done.

L112: "the drainage of the sinking near-surface air" sounds vague to me.
I assumed that the lack of clarity here came from the use of "sinking" and rewrote the sentence as "The local topography controls the drainage of the dense near-surface air as it flows downslope and accelerates […]".

L113: "over an unobstructed"
Corrected accordingly.

L118: "higher incidence of drifting snow", maybe add a reference to Fig. S4?
Done.

L119: "combined with"
Corrected accordingly.

L118-119: I suggest to reference the accompanying Brief Communication here. "Brief communication: Rare ambient saturation during drifting snow occurrences in coastal East Antarctica"
Done.

L160-161: This seems like an appropriate location to introduce Equation 2, instead of Section 3.4. At least, I assume that the same correction is made when part of the FlowCapt was buried as done in Equation 2?
See my response to general comment #4.

L171: Again, I don't think punctual is appropriate here.
The word "punctual" has been replaced with "sporadic".

L194-195: This statement deserves citations.
I refer now to Schmidt (1980) who discusses the importance of inter-particle bonds in the initiation of snow transport and Amory et al. (2017) who discuss the influence of increased threshold friction velocities in summer on snow mass fluxes from data collected at D17 and relate it to the growth of inter-particle bonds.

Fig. 1, as well as Fig. 2: The figure caption should also mention what the gray shaded areas denote (currently it's only explained in the main text).
I have added in the figure caption the meaning of the shaded areas. Note that CRED distributions have been combined into one figure (see fig. R2 and my response to general comment #3) to facilitate comparison between both locations and the figure is now described according to the two main curves only (comments involving shaded areas have been removed) for clarity.

L268: Maybe add: "$h\_ref$ = 2 m, which is the sum of two 1 m long FlowCapt sensors."
Done.

L290: This is confusing. Fig 4, right panel should be a non-log (i.e., linear) scale in order for it to show a linear increase of QT and event duration.

The logarithm scale is preferred here because differences of several orders of magnitude in mass transport per event for all the range of drifting snow events decrease significantly readability when using a linear scale (Fig. R4). I suggest to still make use in the paper of the logarithm scale for readability purposes but include the linear regression fits and their respective equation to highlight the linear character of the relationship between mass transport and duration as shown in Fig. R5.

[Figure]

**Figure R4**. Snow mass transport in drift conditions against duration for each drifting snow event recorded at D47 (red circles) and D17 (blue crosses). Only periods for which two 2G-FlowCapt™ sensors were installed and/or not entirely covered with snow are considered. Linear fits for D47 (black line) and D17 (light blue line) data are also reported on the graph.

[Figure]

**Figure R5**. Logarithm of snow mass transport in drift conditions against duration for each drifting snow event recorded at D47 (red circles) and D17 (blue crosses). Only periods for which two 2G-FlowCapt™ sensors were installed and/or not entirely covered with snow are considered. Linear fits for D47 (black line) and D17 (light blue line) data are also reported on the graph.

L296-297: Could this be substantiated by showing the two levels separately?
Unfortunately the contribution of the saltation and suspension layers to the snow mass flux estimates provided by the 2G-FlowCapt™ cannot be distinguished because fluxes are vertically integrated over the exposed length of the sensor, which for the sensor closest to the ground almost always largely exceeds typical saltation heights (~10 cm) over the measurement period. This has been added to the text.

**References**

Agosta, C., Favier, V., Genthon, C., Gallée, H., Krinner, G., Lenaerts, J. T. M., and van den Broeke, M. R.: A 40-year accumulation dataset for Adélie Land, Antarctica and its application for model validation, Clim. Dynam., 38, 75–86, doi:10.1007/s00382-011-1103-4, 2012.

Baggaley, D. G. and Hanesiak, J. M.: An empirical blowing snow forecast technique for the Canadian arctic and the Prairie provinces, Weather Forecast., 20, 51–62, 2005.

Bintanja, R.: Snowdrift Sublimation in a Katabic Wind Region of the Antarctic Ice Sheet, J. Appl. Meteorol., 40, 15, 2000.

Bromwich, D. H., and Z. Liu, Z.: An observational study of the katabatic wind confluence zone near Siple Coast, West Antarctica. Mon. Wea. Rev., 124, 462–477. 1996.

Cierco, F.-X., Naaim-Bouvet, F. and Bellot, H.: Acoustic sensors for snowdrift measurements: How should they be used for research purposes?, Cold Reg. Sci. Technol., 49, 74–87, doi:10.1016/j.coldregions.2007.01.002, 2007.

Guyomarc'h, G., Bellot, H., Vionnet, V., Naaim-Bouvet, F., Déliot, Y., Fontaine, F., Puglièse, P., Nishimura, K., Durand, Y., and Naaim, M.: A meteorological and blowing snow data set (2000–2016) from a high-elevation alpine site (Col du Lac Blanc, France, 2720 m asl), Earth Syst. Sci. Data, 11, 57–69, 2019.

Mann, G. W., Anderson, P. S. and Mobbs, S. D.: Profile measurements of blowing snow at Halley, Antarctica, J. Geophys. Res., 105(D19), 24491–24508, doi:10.1029/2000JD900247, 2000.

Parish, T. R., and Bromwich, D. H.: Re-examination of the near-surface air flow over the Antarctic continent and implications on atmospheric circulations at high southern latitudes, Mon. Weather Rev., 135, 1961–1973, 2007.

Sato, T., T. Kimura, T. Ishimaru, and T.Maruyama, 1993: Field test of a new snow-particle counter (SPC) system. Ann. Glaciol., 18, 149–154.

Scambos, T. A., Frezzotti, M., Haran, T., Bohlander, J., Lenaerts, J. T. M., Van den Broeke, M. R., Jezek, K., Long, D., Urbini, S., Farness, K., Neumann, T., Albert, M. and Winther, J.-G.: Extent of low-accumulation "wind glaze" areas on the East Antarctic plateau: implications for continental ice mass balance, J. Glaciol., 58, 633–647, doi:10.3189/2012JoG11J232, 2012.

Radok, U.: Snow Drift, J. Glaciol., 19, 123–139, 1977.

Thiery, W., Gorodetskaya, I. V., Bintanja, R., Van Lipzig, N. P. M., Van den Broeke, M. R., Reijmer, C. H., and Kuipers Munneke, P.: Surface and snowdrift sublimation at Princess Elisabeth station, East Antarctica, The Cryosphere, 6, 841–857, doi:10.5194/tc-6- 841-2012, 2012

Trouvilliez, A., Naaim-Bouvet, F., Bellot, H., Genthon, C. and Gallée, H.: Evaluation of FlowCapt acoustic sensor for snowdrift measurements, J. Atmos. Ocean. Technol., 32, 1630–1641, doi:10.1175/JTECH-D-14-00104.1, 2015.

Vionnet, V., Guyomarch, G., Naaim Bouvet, F., Martin, E., Durand, Y., Bellot, H., Bel, C. and Puglièse, P.: Occurrence of blowing snow events at an alpine site over a 10-year period: observations and modelling, Adv. Water Resour., 55, 53–63, doi:10.1016/j.advwatres.2012.05.004, 2013.

---

## Author Comment (AC3) · 3 Feb 2020

**Response to reviewer RC2**

I thank the reviewer for his thorough reading of the paper, the comments and the proposed suggestions. My responses are reported hereafter in red.

This paper presents the analysis of 8 yr field observation of drifting snow at two sites D17 and D47 on Terre Adelie Land (east Antarctica). The main tools used in this study are FlowCapt acoustic sensor and associated Automatic Weather Station. The paper contributes to knowledge concerning the measurement of negative term of surface mass balance driven by wind.

The manuscript subject is appropriate for Cryosphere Journal, well written, data and analysis are very important. The data are partially already presented and analysed in previous paper (Trouvilliez et al, 2014 and 2015) and a paper under review on the Cryosphere (Amory & Kittel, submitted) presents the same data under the aspect of the sublimation that is the main issue not discussed in this manuscript. The interpretations of data acquired are supported by the result and the amount of the good data, but the statistic analysis present in the manuscript are not relevant to support the publication on the high quality Journal such as "The Cryosphere" and the Surface Mass Balance condition from the previous studies are not taken adequately in account. The previous paper on SMB survey at the two sites (D17 and D47) and model on the Adelie Coast are not adequately discussed and reported (example: Pettrè et al., 1986; Bintanja, 1998; Pourchet et al., 1997; Frezzotti el ta., 2004, Genthon et al., 2007; Agosta et al., 2011, Favier et al., 2013, Barral et al., 2014 in the reference but is not taken in account in the manuscript; Goursaud et al., 2017). The manuscript does not analysed the Surface Mass Balance and in particular the extensive presence of the blue ice area in the Coastal Terra Adélie (Favier et al., 2011) and their implication on the drifting snow.

The author does not distinguish drift from blowing snow phenomena and the threshold of snow sublimation, and their implication on the mass transport/sublimation and the difference between the two sites. Blowing and drifting snow are not redistribution process, a significant part of blowing snow sublimate as pointed out by snow radar survey (see Frezzotti et al., 2007; Eisen et al., 2008) or satellite survey (Scarchilli et al., 2010, Palm et al, 2011, 2017; Scambos et al., 2012). The AWS and FlowCapt sensor provide single measurement point for limited number of years and must be analysed in the contest of Surface Mass Balance study derived from other field measurements as stakes, firn cores, snow radar profile and satellite studies.

Although rather concise, many issues are raised in this report and I'll try to address all of them. Note that some of the elements of response provided here are redundant with the information provided in the manuscript but are reported here since they constitute key elements of the argumentation.

One major issue raised by the reviewer is the lacking character of the publication concerning surface mass balance aspects, and the discrepancy between the reviewer's expectations and the content of the paper. If the subject of the paper were to investigate the relations between erosion and variability in the surface mass balance in Adelie Land, I should have indeed considered those points raised by the reviewer, I agree, although one would face some serious complications by doing so, as I will discuss it in the following paragraph. But the objective of the paper in the proposed version is different: I aim here at publishing and presenting the drifting snow database while providing some examples of use of the data through a first statistical (temporal and spatial) analysis of drifting snow mass transport and frequency (made possible by the high sampling frequency and the continuous character over the respective measurement periods) and in which emphasis is placed on aspects relevant to the modelling of drifting snow. The interest of the paper partly relies on the quality and the open-access character of the drifting snow database (which has been deposited on zenodo and can be now downloaded at https://zenodo.org/record/3630497; see my response to reviewer RC1), that compiles new observations which are almost inexistent in the extreme and remote Antarctic environment. By making them freely available to the scientific community without condition, the paper is an opening to a larger field of

applications, such as evaluation of climate models, simultaneous analysis of ground-based and remotely sensed data, investigations on polar boundary-layer physics or accumulation/ablation processes, each of them belonging to a specific area of expertise and individually warranting a careful, detailed, equally interesting attention.

It is of crucial importance to understand the possibilities offered by the drifting snow data. In particular, the relation between the snow mass flux and ablation at a given area is far from being direct. Snow mass fluxes do not constitute an estimation of local erosion; rather they are the integrated result of all the mechanisms that contribute to the presence, amount and time residence of snow particles in the air, including notably precipitation and advection from upwind areas. Unfortunately the FlowCapt™ sensor does not distinguish neither the source or the geographical origin of the particles impacting the tube, and how much precipitation contribute to the snow mass transport in Antarctica is still an open research question. Therefore these observations must not be perceived as measurements of "wind-driven negative term of the surface mass balance" and cannot be used to quantify local ablation rates without the use of a complementary approach such as numerical modelling. This is the main reason why all the references mentioned by the reviewer are indeed not discussed in the paper since they would lie beyond the scope of the paper. Some of other major concerns that would arise from the multidisciplinary and quite ambitious approach suggested by the reviewer result from the fact that:

- the transect along which stake measurements are indeed performed in Adelie Land is not aligned with the main slope in the wind direction, so any spatial variability depicted in the local SMB signal might not directly correlate to the magnitude of the snow mass flux.

- the fine spatial resolution of the stake networks in Adelie Land demonstrates a high, sub-kilometre spatial variability that couldn't be supported by analysis of mass fluxes performed at only two distant locations, and whose spatial representativeness cannot be assessed in the absence of other comparable measurements in the area.

- the observations presented in this paper are, as you mentioned, "*single measurements point for limited number of years*", moreover performed over the most recent years. A mismatch in timing of several decades can thus be expected with the information contained in deep snow/ice layers sampled through ice cores and radar stratigraphy.

Moreover, temporal and spatial variability of the SMB in Adelie Land has already been quite well investigated (as demonstrated by the long, non-exhaustive list of references mentioned in your report), while drifting snow mass transport has received much less attention. Documentation of spatial et temporal variability in snow mass transport over Antarctica almost exclusively relies on models whose ability to represent drifting snow processes has been extremely limited owing notably to the current extreme scarcity of drifting snow observations. This justifies, in my humble opinion, an initial and independent documentation of the entire database before exploiting further possible connections with other processes and synergetic uses with other products. For the above-mentioned reasons, and because I'm also deeply convinced that keeping the scientific message of a paper as onefold improves clarity, readability and efficiency and thus prevent the paper from being too long with various scientific messages and disconnected sections, I believe that the various applications mentioned by the reviewer are all interesting subjects for separate papers. Finally, note that SMB-related aspects are discussed in a manuscript that I've submitted to Geoscientific Model Development, in which the data presented in this paper have been used in conjunction with the SMB observations in Adelie Land to evaluate the drifting snow scheme of the regional climate model MAR and its ability to represent the variability in accumulation along the transect.

The following comments gather some elements of responses to the remaining comments of the reviewer:

- The driting snow observations are performed far from the blue ice area (i.e. respectively 10 and 100 km away from the ice margin), which covers the very first hundreds of metres of the icesheet margin, and relate to different local (topographical) conditions irrelevant to the objective of the paper in its current form.

- As the sensors are installed at both measurement sites so as to cover the first two meters above ground, which is the height conventionally used to distinguish drifting from blowing snow, they do not enable a distinction between drifting and blowing snow.

- Sublimation of windborne snow is determined by the temperature and humidity gradients across the boundary layer between each snow particle and its environment and is proportional to the undersaturation of the atmosphere (Schmidt, 1982). For a given wind speed, threshold values at which airborne snow sublimation becomes significant can thus be expected to vary significantly depending, among others, on the snow particle concentration and thermodynamic properties and structure of the atmosphere, or the dynamical origin of the boundary-layer flow. From this perspective, as a *"threshold of snow sublimation"* sounds quite vague to me, I also assumed that the reviewer possibly meant "wind speed thresholds for snow transport". A comparison between both measurements sites in terms of occurrence of drifting snow as a function of wind speed is already proposed and discussed in the paper (see Figs. 2 and 3 in the former and revised version of the manuscript, respectively). Moreover, the actual quantity involved in the triggering of drifting snow is the friction velocity, which is only dependent on surface snow properties (Gallée et al., 2001). In the absence of measurements of surface snow properties, and knowing that atmospheric flow conditions would also influence the results, accurate determination of such thresholds can be achieved through turbulence measurements (not available either at D17 or D47) or wind speed profiles using the similarity theory by computing the friction velocity at the onset of drifting snow (e.g., Trouvilliez et al. 2014). However, such an alternative involve a thorough determination protocole (Amory et al., 2017) and selection criteria that are not continuously and homogeneously met at D17 and would result in a discontinuous time series, additionally subject to impeding factors (see my response to general comment #3 of reviewer RC1). For instance, accuracy issues arising from various variable numbers of available anemometers, absence of knowledge of measurement heights, choice of stability correction functions, or the validity of the similarity theory in drifting snow conditions, as well as inclusion of drag effects to the shear stress estimates (e.g., Amory et al., 2016) and artificial roughness created during maintenance operations, are all arguments that would, again, certainly deserve an entire sensitivity study in the form of another publication. Finally, the single measurement level at D47 preclude such determinations, and therefore spatial comparison with D17.

- Limitations in computing sublimation rates from AWS data and drifting snow mass fluxes have already been discussed in a recently published paper (Amory and Kittel 2019). The authors made use of one year of this dataset at site D17 in complement to relative humidity profiles to investigate the development of a near-saturated surface air layer in relation to the occurrence of drifting snow. As also mentioned in that paper, such an exercise involves specific requirements that are only met during a reduced period of time at site D17, justifying its treatment in a separate publication and precluding its application to D47 and outside of the period of study (year 2013). Similarly, strong limitations in the use of the thermo-hygrometers and in the applicability of the Monin-Obukhov similarity theory for retrieving latent heat fluxes at D17 (from which drifting snow sublimation rates could be inferred) have also been discussed in Barral et al. (2014).

- Sublimation of windborne snow has been inferred from accumulation measurements (e.g. Frezzotti et al., 2007; Scambos et al., 2012) but still remained to be confirmed and quantified by measurements of the latent heat flux within the atmosphere and drifting/blowing snow layers, accounting for the physical constraints mentioned above. The usual alternative is the use of gridded model products. Attempts using also satellite data have been made (Palm et al., 2017), but they involve the use of (i) parameterizations for snow particles properties, (ii) snapshots of the atmospheric conditions that are representative of instantaneous conditions only, and (iii) reanalysis produced from model that do not take into account interactions of snow particles with the atmosphere, particularly the negative feedback of windborne snow sublimation thus leading

to a dry bias that can result in strong overestimation of sublimation rates and give the role of an infinite mass sink to the atmosphere. Moreover, as discussed in the introduction, strong discrepancy currently remain between the available model products (~100 vs 400 Gt/an), to the extent that the difference between each estimate is one order of magnitude higher than any other ablation term of the surface mass balance as determined from regional models (e.g., Agosta et al., 2019; Mottram et al., 2020). Different model-based approaches have also been proposed in which drifting snow mass transport is believed to be the first-order process with respect to sublimation, because of the low capacity of the atmosphere to hold moisture in the cold environment where accumulation measurements have been performed (Agosta et al., 2019). The role of sublimation during snow transport, particularly as a negative ablation term in Antarctica, is a currently debated problem in meteorology and snow science. I kindly refer to Amory and Kittel (2019) for a more detailed discussion on that matter from D17 data.

- Note that in Trouvilliez et al. (2014) only the initial results of the drifting snow observation campaign (the first 2 years) are presented under different processing criteria relative to a less complete knowledge on the FlowCapt™ capabilities at the time of redaction, and with no accessibility of the data and much less emphasis on drifting snow mass transport.

- Trouvilliez et al. (2015) focus on data collected in the French Alps and their work is not connected to the observations in Adelie Land.

**References**

Amory, C., Gallée, H., Naaim-Bouvet, F., Favier, V., Vignon, E., Picard, G., Trouvilliez, A., Piard, L., Genthon, C. and Bellot, H.: Seasonal variations in drag coefficient over a sastrugi-covered snowfield in coastal East Antarctica, Bound.-Lay. Meteorol., 164, 107–133, 2017.

Amory, C. and Kittel, C.: Brief communication: Rare ambient saturation during drifting snow occurrences at a coastal location of East Antarctica, The Cryosphere, 13, 3405–3412, https://doi.org/10.5194/tc-13-3405-2019, 2019.

Amory, C., Naaim-Bouvet, F., Gallée, H. and Vignon, E.: Brief communication: two well-marked cases of aerodynamic adjustment of sastrugi. The Cryosphere 10(2),743–750, doi:10.5194/tc-10-743-2016, 2016.

Barral, H., Genthon, C., Trouvilliez, A., Brun, C., and Amory, C.: Blowing snow in coastal Adélie Land, Antarctica: three atmospheric-moisture issues, The Cryosphere, 8, 1905–1919, https://doi.org/10.5194/tc-8-1905-2014, 2014.

Frezzotti, M., Urbini, S., Proposito, M., Scarchilli, C., and Gandolfi, S.: Spatial and temporal variability of surface mass balance near Talos Dome, East Antarctica, J. Geophys. Res., 112, F02032, https://doi.org/10.1029/2006JF000638, 2007.

Gallée, H., Guyomarc'h, G., and Brun, É.: Impact of snow drift on the antarctic ice sheet surface mass balance: possible sensitivity to snow-surface properties, Bound.-Layer Meteorol., 99, 1–19, 2001.

Mottram, R., Hansen, N., Kittel, C., van Wessem, M., Agosta, C., Amory, C., Boberg, F., van de Berg, W. J., Fettweis, X., Gossart, A., van Lipzig, N. P. M., van Meijgaard, E., Orr, A., Phillips, T., Webster, S., Simonsen, S. B., and Souverijns, N.: What is the Surface Mass Balance of Antarctica? An Intercomparison of Regional Climate Model Estimates, The Cryosphere Discuss., https://doi.org/10.5194/tc-2019-333, in review, 2020.

Palm, S. P., Kayetha, V., Yang, Y. and Pauly, R.: Blowing snow sublimation and transport over Antarctica from 11 years of CALIPSO observations, The Cryosphere, 11, 2555–2569, doi:10.5194/tc-11-2555-2017, 2017.

Scambos, T. A., Frezzotti, M., Haran, T., Bohlander, J., Lenaerts, J. T. M., Van den Broeke, M. R., Jezek, K., Long, D., Urbini, S., Farness, K., Neumann, T., Albert, M. and Winther, J.-G.: Extent of low-accumulation "wind glaze" areas on the East Antarctic plateau: implications for continental ice mass balance, J. Glaciol., 58, 633–647, doi:10.3189/2012JoG11J232, 2012.

Schmidt, R. A.: Vertical profiles of wind speed, snow concentration, and humidity in blowing snow, Bound.-Lay. Meteorol., 23(2), 223–246, doi:10.1007/BF00123299, 1982.

Trouvilliez, A., Naaim-Bouvet, F., Genthon, C., Piard, L., Favier, V., Bellot, H., Agosta, C., Palerme, C., Amory, C. and Gallée, H.: A novel experimental study of aeolian snow transport in Adelie Land (Antarctica), Cold Reg. Sci. Technol., 108, 125–138, doi:10.1016/j.coldregions.2014.09.005, 2014.

Trouvilliez, A., Naaim-Bouvet, F., Bellot, H., Genthon, C. and Gallée, H.: Evaluation of FlowCapt acoustic sensor for snowdrift measurements, J. Atmos. Ocean. Technol., 32, 1630–1641, doi:10.1175/JTECH-D-14-00104.1, 2015.

---

## Author Comment (AC4) · 12 Feb 2020

Dear Editor,

Please find below our point-by-point reply to each reviewer's comment, followed by a marked-up manuscript version explicitly showing the changes made compared to the original version.

More specifically, please note that, as requested by reviewer RC1 (i) in the revised manuscript section 2 (description of drifting snow data) has been expanded and includes a more detailed assessment of the FlowCapt sensors from new data collected in Adelie Land, and (ii) the database described and presented in the paper has been deposited in a public repository via zenodo and is now freely available. Statistics have also been slightly altered due to the redefinition and refinement of some selection criteria based on RC1's comments and have been adjusted accordingly. However, please note that no change has been made according to the short report of reviewer RC2 as it mainly relates to applications that lie, to my opinion, beyond the scope of the paper and have moreover been treated in another manuscript under revision.

Kind regards

Charles Amory

I sincerely thank the reviewer for its thorough reading of the paper, the very relevant questions and the interesting suggestions that will undoubtedly help to improve the paper. My responses are reported hereafter in red.

**Response to reviewer RC1**

The manuscript describes a unique long dataset of blowing snow observations in Adelie Land, Antarctica, using weather stations that contained FlowCapt sensors for detecting blowing and drifting snow particles. Given that the surface mass balance in Antarctica is dominated by wind erosion and deposition (and thus blowing snow), in-situ measurements are very valuable. It's important that the manuscript is published, describing the setup, and providing some first analysis of frequency of blowing snow events, to increase the value of the dataset. The dataset may become useful for a broad community, including remote sensing and ice sheet (climate) modelling. I think the paper is well suited for TC. I have some minor comments which may help to improve the paper. A major concern I have is that the data is only available from the author on request, whereas I think that the Copernicus Data Policy strongly discourages this. Using for example zenodo and doi versioning, the current dataset could be deposited there, and updated with newer data as a newer version of the data.

The data has been deposited on a public repository via zenodo and can now be downloaded at https://zenodo.org/record/3630497. The section data availability has been modified accordingly.

Please note that the SPC data used for the evaluation of FlowCapt sensors as proposed in the revised version of the manuscript are currently involved in an ongoing publication and have thus not been deposited together with the drifting snow dataset on zenodo.

**General comments**

1) A better discussion of the accuracy of FlowCapt is required. In section 2.3, L140, the sensor is described as being accurate, and the only reference is Cierco et al. 2017). However, they write: "As a consequence, and even if the sensor provides good information in operational use, a regrettable inaccuracy in the collected data prevents the use of such measurements for research purposes. Nevertheless, a correction algorithm based on a statistical calibration of the sensor is proposed, which should make it possible to use the recorded data for preliminary approximations." In L151-153, it is discussed qualitatively, but can quantitative error margins be established? In any case, this section needs to be expanded, with a more detailed accuracy assessment of FlowCapt sensors.

All the work done by Cierco et al. (2007) on evaluating the reliability of the Flowcapt[TM] sensor and characterizing its limitations has focused on the first-generation device. The sensors installed at D17 and D47 are of a more recent design (referred to as second-generation FlowCapt or 2G-FlowCapt in the paper) which significantly improves, without necessarily solving, inaccuracy issues with estimating snow mass fluxes (Trouvilliez et al., 2015).

I have reorganized Section 2.3 to better structure the information provided on 2G-FlowCapt[TM] sensors and utilization of the data. The section is now divided in four parts whose one of them is dedicated to accuracy assessment of the sensors. More specifically I mention results (which I suggest to provide in supplementary materials with further details on the experimental set-up and methodology) from a intercomparison experiment I've led myself in Adelie Land between the 2G-FlowCapt[TM] and a snow particle counter (SPC-S7), an optical device considered as a kind of reference sensor for estimating snow mass fluxes (Sato et al., 1993). The field experiment took place at site D17 in late January 2014 and focused on a 24 hour long snow transport event. Although more data are necessary to better assess the performance of the 2G-FlowCapt[TM] in Antarctic conditions, the comparison shows a reasonable agreement between the two types of sensors (Fig. R1; proposed as Fig. S4 in supplementary materials). Reported below is the content of Sect. 2.3 in the revised version of the manuscript entitled "Accuracy assessment" and the related supplementary materials; please refer to the track changes for a complete report of the modifications undertaken in this section:

*"[…].*

**2.3.3. Accuracy assessment**

*While FlowCapts™ sensors can detect the occurrence of snow transport with a high level of confidence, the ability of the original design to estimate snow mass fluxes is more questionable (Cierco et al., 2007). These accuracy issues, without being necessarily solved, have been significantly improved with the 2G-FlowCapt™, facilitating its use for quantitative applications (Trouvilliez et al. 2015). Although measurement uncertainty is not known, the 2G-FlowCapt™ was shown to generally underestimates the snow mass flux relatively to integrated estimates computed from optical measurements made with a snow particle counter S7 (SPC-S7; taken as a reference in the study) during a winter season in the French Alps, particularly during concurrent precipitation (Trouvilliez et al. 2015). During mixed drifting snow events when erosion occurs simultaneously with snowfall, the density of precipitating particles which have not reached the ground yet is lower than eroded, more rounded snow particles originating from the ground which have lost their original crystal shape and size through collision, sublimation and the thermal processes of metamorphism. For a given snow mass flux, the particles' momentum, and by extension the measured acoustic pressure, is therefore lower during a mixed drifting snow event than during an event predominantly driven by the erosion process. This results in an underestimation of the snow mass flux measured by the 2G-FlowCapt™ during mixed events, with a magnitude depending on the relative proportion of eroded particles against fresh snow particles.*

*Environmental conditions influence greatly the estimation of the snow mass flux by the 2G-FlowCapt™. The intercomparison experiment in the Alps was done within a range of mass flux values ($< 2.5 \ 10^{-2} \ kg \ m^{-2} \ s^{-1}$) significantly lower than those encountered in Adelie Land (see Sect. 3.4). In addition, comparatively stronger surface winds and lower temperature on the Antarctic ice sheet favor the breaking and rounding of snow particles. This suggests that the performance of the 2G-FlowCapt™ remains to be assessed in the extreme Antarctic environment, in which large proportions of small, rounded particles can be expected in drift conditions (i.e. within 2 m above ground) even with concurrent precipitating snow (Nishimura and Nemoto, 2005).*

*A field experiment involving measurements with SPC-S7 and 2G-FlowCapt™ sensors performed during a 24 hour long snow transport event was undertaken at site D17 in late January 2014 (Fig. S3). Strong drift conditions were observed with 2-m wind speeds and snow mass fluxes reaching up to 19 m s⁻¹ and $4 \ 10^{-1} \ kg \ m^{-2} \ s^{-1}$ respectively. Although the statistical representativeness of the results may be small due to the low amount of data collected during only one event, the comparison shows that the snow mass fluxes provided by the two types of sensors are very similar in magnitude (Fig. S4). Further details on the experimental set-up and comparison methodology are provided in supplementary materials (Sect. S1).*

[Figure]

**Figure R1**. Comparison between snow mass fluxes provided by 2G-Flowcapt™ sensors and computed from measurements made with snow particle counters (SPC-S7) during a snow transport event at site D17 in January 2014. A distinction is made between snow mass fluxes integrated over 0.1 to 1.1 m and 1.2 to 2.2 m above ground.

Supplementary materials:

**S1. Intercomparison between snow particle counters S7 and second-generation FlowCapt™ sensors during a drifting snow event in Adelie Land**

**1.1. Snow particle counters**

The measurement principle of the snow particle counter S7 (SPC-S7) follows an optical method based on the strong absorption of the infrared light by the snow. The diameter and number flux of snow particles are detected by their shadows on a super-luminescent diode sensor. Electric pulse signals corresponding to a snow particle passing through a sampling area of 50 mm² (2 mm in height and 25 mm in width) and whose voltage is directly proportional to the size of the particle are classified into 32 size bins from ~ 40 to 500 μm (Sato et al., 1993). This means that snow particles smaller than 40 μm remain undetected and snow particles larger than 500 μm are assigned to the maximum diameter class. Thanks to a self-steering vane the SPC-S7 measures perpendicularly to the horizontal wind vector the distribution size spectrum of snow particles every 1 s, from which the horizontal snow mass flux, η, can be computed assuming fully spherical snow particles with a density equal to that of ice as follows:

$$\eta = \sum_{i_d=1}^{32} \eta_d = \sum_{i_d=1}^{32} n_d \frac{4}{3} \pi \left(\frac{d}{2}\right)^3 \rho_i$$

with $\eta_d$ (kg m⁻² s⁻¹) the horizontal snow mass flux for the class of diameter d (m), $i_d$ the index and $n_d$ the measured number flux of snow particles (part. m⁻² s⁻¹) for each of the 32 diameter classes, and $\rho_i$ the particles density (917 kg m⁻³).

**1.2. Experimental set-up**

Two SPCs were installed on 28 January 2014 (Fig. S3) a few hours before strong drifting snow occurred in conjunction with strong katabatic winds reinforced by the passage of a low-pressure system off the Adelie Coast. The equipment was removed on 29 January once drifting snow ceased. One SPC was installed at a fixed position 1 m above the ground, while the position of the other was alternatively switched manually between 0.5 et 2 m above the ground every 1-2 hours. This was done in order to study the vertical gradient of the mass flux for two ranges of height (0.1-1.1 m and 1.2-2.2 m) above the snow surface for which 2G-FlowCapt™ measurements are also available for comparison. The high energy requirements of the SPCs (~ 15 W) were fulfilled by an electric generator that was housed together with the acquisition system in a mobile shelter downwind of the measurement structure. Only a few data are missing due to problem with the acquisition system of the SPC at the beginning of the experiment, resulting in an timeseries almost continuous along the event.

**1.3. Computation of integrated snow mass fluxes from SPC data**

According to the diffusion theory of drifting snow (Radok, 1977), the averaged drifting snow particle density (kg m⁻³) in the diffusion layer can be approximated by a function of height. When the wind profile follows a power law, an expression for the vertical distribution of the snow mass flux η(z) (kg m⁻² s⁻¹) writes

$$\eta(z) = az^{-b}$$

where a is the calibration parameter and b the exponent independent of height. These parameters were derived by regression from the data measured by the two SPCs (Trouvilliez et al. 2015), alternatively available for the two height ranges. Then, the half-hourly average of the horizontal snow mass flux vertically integrated over the corresponding height covered by the 2G-FlowCapt™ can be estimated. Because (i) snow depth measurements revealed insignificant height change after the event and were affected by the presence of drifting snow particles perturbing the travel of ultrasound pulses along the measuring path during the event, (ii) the two 2G-Flowcapts were respectively installed at 0.1 and 1.2 m above the snow surface at the beginning of the event, and (iii) the heights of the SPCs were regularly checked and manually adjusted along the experiment, constant heights are used in the integration. Finally, data were processed following the procedure described in Guyomarc'h et al. (2019).

Resulting integrated snow mass fluxes are compared in Fig. S4. Although more data are necessary to better assess the performance of the 2G-FlowCapt$^{TM}$ in Antarctic conditions, a high degree of agreement between the two types of sensor is depicted with a correlation coefficient of 0.82 and 0.93 and a rmse of 70 10⁻² and 13 10⁻² kg m⁻² s⁻¹ (by taking the SPC-S7 as a reference) for the lower and upper measurement range, respectively.

[Figure]

*Figure S3. Picture of the snow particle counters installed at D17 during the intercomparison experiment in late January 2014.*

*Figure S4. See Fig. R1.*

2) It is not clear what the measurement heights were, particularly for D17. Also, it should be made clear throughout the manuscript which measurement level is used for wind speed, temperature and RH, at D17 when plotting or analyzing.

At D17 I've selected the measurement height closest to 2-m as recovered by the snow depth ranger or using the closest original height when no information on snow height is available. This is now explicitly detailed in the manuscript and specified in required places.

3) Fig. 2 shows that drifting snow occurs with lower wind speeds in D17 than in D47. Is this because the surface snow density, or bond strength is lower at this site? Lower average wind speeds can be associated with lower surface density. Or are the two measurement heights not comparable? It would make sense to also discuss the occurrence of drifting snow as a function of friction velocity, as mentioned in L238-239 which is often the variable of interest, determining whether or not snow erosion from the surface occurs. For D17, the profile measurements easily allow for a determination of friction velocity from the 6 measurement heights of wind speed. Similarly, L288, surface roughness could also be quantified from the wind speed profile, which could be used to demonstrate this relationship.

The definition of a drifting snow occurrence in the former version of the manuscript involved a drifting snow mass flux above the confidence threshold at either measurement level. After some reconsideration motivated by your comment on the lack of clarity about the definition of a drifting snow event, I figured out that this may still generate few artificial occurrences when fluxes at the upper level are above the threshold value while fluxes at the lower one are not for small values oscillating around the threshold value. I have then slightly modified the definition of a drifting snow occurrence and redone the analysis with the following definition:

"[…] *drifting snow has been considered to occur when the half-hourly mean of the snow mass flux exceeds a confidence threshold of $10^{-3}$ kg m$^{-2}$ s$^{-1}$ as determined from visual observations on the field in Adelie Land (Amory et al., 2017). Note that the same confidence threshold yielded a high level of agreement (98.6 %) between the SPC-S7 and 2G-FlowCapt$^{TM}$ in terms of occurrence detection in the comparison study led by Trouvilliez et al. (2015) in the Alps. Since this value remains small compared to snow mass fluxes estimated during drifting snow occurrences (see Sect. 3.4), the confidence threshold is assumed independent on the exposed length of the sensor.*

*The sensor is considered unburied as long as at least 10 % (i.e. 0.1 m) of its initial length remain uncovered with snow."*

The resulting CRED distribution is presented in Fig. R2. Note that this figure is proposed in replacement of Fig. 2 in the revised version of the manuscript.

Indeed lower surface densities that can be expected at D17 due to lower wind speeds could enable drifting snow to occur at lower wind speeds than at D47. Conversely lower drift-induced compaction at D17 could be compensated by stronger interparticle bonding resulting from higher average temperatures. There is unfortunately no direct way to investigate that aspect, and answering your question in a more exhaustive way would inexorably require knowledge of snow properties at the surface, or at least determination of threshold friction velocities at D47, which are either not available or possible due to the single measurement level at D47. I then suggest to leave this question open for further studies. Anyways, updated Fig. R2 no longer suggests that such processes may be responsible for significant differences in CRED distribution between the two locations despite the actual differences in climate conditions, and indicates that wind speed is the main deriver behind the occurrence of drifting snow.

Accurate determination of friction velocity values from the data collected in the extreme environment of D17 involve several selection criteria (see Amory et al. 2017) that are far from being systematically met along the measurement period, resulting in discontinuous time series that can be additionally biased by an over-representation of climate conditions for which these criteria are met but not necessarily representative of the average climate conditions. In particular, the validity of Monin-Obukhov similarity theory, from which friction velocity and roughness length can be determined using the wind speed profiles, is questionable during drifting snow conditions. Another influent factor in the determination of friction velocity that could impede the generation of homogeneous time series is the number of anemometers available for each wind profile, which varies non-uniformly along the measurement period depending on instrument failure and burial of the lower levels. Note that evolution of the measurement height is also not known before 2013 at site D17. These factors, together with additional parameters such as the choice of the stability correction function, all justify a sensitivity study that lies beyond the scope of the present paper. A non-negligible issue when using the wind profiles at D17 to compute friction velocity values is the artificial roughness created during summer visits and maintenance/replacement operations. This is especially true since summer 2014-2015 when the meteorological mast at D17 was mounted on a sledge that was further buried under the snow. The whole operation, ideally repeated each year, requires excavation of several meters in depth within the snowpack and use of snow trucks that disturb surface roughness in the immediate vicinity of site D17 for an unknown duration.

For all these reasons, and also because site D47 is equipped with only one measurement level thus precluding spatial comparison with D17, I chose to focus on wind speed rather than friction velocity, enabling the discussion of continuous, more homogeneous times series at both locations simultaneously.

Similarly, threshold friction velocity would need to be continuously known all along each drifting snow event to investigate differences with friction velocity, so the theoretical statement made in L288 cannot be illustrated with the present data. It has therefore been removed from the text (see my response to the next comment). Here in the absence of direct measurements of surface snow properties, threshold friction velocity values can only be determined at the onset and end of drifting snow event, when the snow mass flux rises or drops below the confidence threshold value and only if friction velocity can be accurately determined from the wind speed profiles at the same time.

[Figure]

**Figure R2.** CRED distribution for drifting snow occurrences showing the increasing probability of observing drifting snow with increasing 2-m wind speed at sites D47 (red curve) and D17 (blue curve). Shaded areas correspond to CREDs respectively computed using a relaxed and a stricter confidence threshold of $10^{-4}$ kg m$^{-2}$ s$^{-1}$ and $10^{-2}$ kg m$^{-2}$ s$^{-1}$ and are shown as a measure of uncertainty.

4) Eq. 2 confuses me, because the summation symbol lacks what it is summing over. At first, I thought that this was just how to compute half-hourly transported mass, but it seems to be for event transported mass and that the summation is over all the time steps constituting an event. In that case, L276 should introduce the definition of drifting snow event, if it is not each half-hourly data interval. Is the first half hourly interval that drops below the limit considered the end of the event? Nowhere in L262-271 is it introduced that there is a switch to the event based analysis.

Several modifications have been made in order to improve clarity when dealing with the computation of snow mass fluxes and snow mass transport and are reported below:

(i) Derivation of snow mass transport from former Eq. 2 has been re-expressed in a clearer way following two equations, starting from the formulation of the drifting snow mass flux $\eta_{DR}$ (i.e. vertically integrated between 0 and 2 m over the snow surface) as

$$\eta_{DR} = \begin{cases} \eta_1 + \eta_2, & h_1 + h_2 \geq h_{ref} \\ \eta_1 + \eta_2 \cdot \dfrac{h_{ref}}{h_1 + h_2}, & h_1 + h_2 < h_{ref} \end{cases} \tag{1}$$

*where $\eta_i$ (kg m$^{-2}$ s$^{-1}$) is the observed snow mass flux integrated over the exposed height $h_i$ (m) of the corresponding 2G-FlowCapt™ sensor, and $h_{ref} = 2$ m is the sum of two fully exposed, 1 m long 2G-FlowCapt™ sensors. In other words, when $h_1 + h_2 < 2$ m, it is assumed that the measured snow mass flux is constant up to 2 m. To keep consistency with the confidence threshold for the detection of drifting snow occurrences, snow mass fluxes below $10^{-3}$ kg m$^{-2}$ s$^{-1}$ have been set to 0. The horizontal drifting snow mass transport for a given period of time $[t_0, t_n]$, $Q_{DR}$, then writes*

$$Q_{DR}(t) = \int_{t_0}^{t_n} \eta_{DR}(t)dt. \tag{2}$$

(ii) Following your recommendation, Eqs. 1 and 2 are now introduced in the methods section,

(iii) The definition of a drifting snow event in Sect. 3 has been made clearer and is now presented as "*a period over which snow transport is detected for a minimum duration of 4 hours. That is, an event is considered to start and end when the half-hourly snow mass flux at the lower unburied level $\eta_i$ respectively rises and drops below the confidence threshold of $10^{-3}$ kg m$^{-2}$ s$^{-1}$.*",

(iv) Complementary information has been given on the threshold used for acknowledging the occurrence of drifting snow in Sect. 2.3 as detailed in my response to general comment #3,

(v) All along the manuscript it is now explicitly mentioned to which snow mass flux (raw sensor output $\eta_i$ or corrected drifting snow mass fluxes $\eta_{DR}$) I refer to.

The relation between snow mass transport and wind speed is now discussed by studying half-hourly drifting snow mass fluxes computed from Eq. (1) as a function of related 2-m wind speeds (Fig. R3 – Fig. 5 in the revised version of the manuscript) instead of mass transport and average wind speed per event, and consists in the following analysis in the paper:

"*The drifting snow mass flux $\eta_{DR}$ typically tends to increase with wind speed in a power-law fashion (Fig. 5). This well-known behavior (Radok, 1977; Mann et al., 2000) is however depicted with significant dispersion and notable differences between the two locations; the data at D17 show that drifting snow mass fluxes can be of greater magnitude than at D47 for similar wind speeds and exhibit a generally higher variability along the range of wind speeds. This illustrates the diversity and spatial variability in factors controlling the windborne snow mass, as mentioned in the previous section. While wind speed can be used to predict the occurrence of drifting snow with a quite similar probability distribution between both locations (Fig. 3), on the other hand Fig. 5 demonstrates that more caution should be taken when scaling drifting snow mass transport with wind speed or related single parameter independent of surface snow properties (e.g. Mann et al., 2000). Such an approach would indeed involve mixtures of power laws to capture the large variability in drifting snow mass flux within the same wind speed interval, particularly at D17 where almost the entire range of values is observed from 15 m s$^{-1}$. Drifting snow is highly non-linear in nature and results essentially from the competitive balance between atmospheric drag and cohesive forces acting on the snow surface. This means that concurrent documentation of turbulence and surface snow properties are required for a better assessment of drifting snow processes and improvements of model predictability (e.g. Baggaley and Hanesiak, 2005; Vionnet et al., 2013).*".

[Figure]

**Figure R3**. Drifting snow mass flux against 2-m wind speed recorded at D47 (left panel) and D17 (right panel). Only periods for which two 2G-FlowCapt™ sensors were installed and/or the lower sensor was not entirely covered with snow (i.e. $h_1 > 0.1$ m) are considered.

As the definition of a drifting snow event is explicitly given in Section. 3.3, make a recall just a few paragraphs below would thus sound redundant. The paragraph describing the relation between mass transport and event duration introduces the switch to the event-based the analysis in the 2$^{nd}$ sentence: "*Values of $Q_{DR}$ have been computed for each drifting snow event identified in the database [...]*".

5) Cierco et al. (2007) mentions that FlowCapt sensors can saturate. Could that be an explanation for the apparent upper bound on snow transport mass (L291-292)?

If saturation of the sensors would be responsible for the apparent upper bound in snow mass transport, it should be apparent from the raw sensor outputs as well. However no evidence of such a behavior is found when plotting the half-hourly snow mass fluxes against wind speed as proposed in the new analysis (see the previous comment - Fig. R3). Note also that the evaluation of Cierco et al. (2007) makes use of the original design of the FlowCapt sensor, and the analysis proposed here relies on a more recent design which significantly improves, although not necessarily solve, inaccuracy issues with estimating snow mass fluxes (Trouvilliez et al., 2015).

6) I'm not sure if the Online Supplement is necessary. It seems that the manuscript would benefit from inclusion of most of the materials in the main text. In particular, I think that the map with location of the stations should be part of the main text.
Yes I agree, the location of the measurement sites is of first importance. The map has been included in the main text. I'd like to stress that I have a limited budget for this publication and inclusion of figures in the main text rather than in supplement goes with significant charges. So I'd like to stick to the very essential material in the main text if it seems reasonable, and leave the rest (Tables S1,S2 and Figs. S2,S2) as supplementary materials since they contain additional information that would surely be beneficial but are not really essential to the analysis.

**Minor comments**
L7: "hydrological"? Maybe "surface mass balance" is better (see L21-22)?
Thanks for the suggestion. Surface mass balance is the resultant of the hydrological cycle at the surface of the ice sheet, so I'd rather keep this formulation as it is, if it does not result in any misunderstanding.

L8: "punctual" doesn't seem to be the right word here. I suggest: "model and satellite based products".
I simply removed "punctual" and now speak about "model and satellite products".

L13: "The data provided nearly continuously so far constitutes". Maybe add commas for clarification: "The data, provided nearly continuously so far, constitutes"
Corrected accordingly.

L25: It's confusing: wind confluence for me is the *convergence* of wind, unless there is compensating acceleration (which doesn't seem to be the case given that D47 has higher wind speeds than D17), so how does this relate to the horizontal *divergence* of snow?
I agree that these two terms might be confusing when employed together as they may apparently relate to opposite situations. In a general picture, convergence of the wind field is associated with deceleration and thus deposition of snow (or convergence of snow by wind transport). While this is certainly verified over a flat surface, this does not however preclude the occurrence of erosion within steep regions of the confluence area, where katabatic winds converge from a large-scale perspective and still can accelerate locally depending on the slope of the ice surface.
Changes in surface slope play a crucial role in controlling locally the erosion/deposition process through acceleration/deceleration of the near-surface flow. This is illustrated through sub-kilometer spatial heterogeneities in accumulation in, for instance, coastal Adelie Land (Agosta et al., 2012), or alternating ridges and glazed surfaces scattered over East Antarctica (Scambos et al., 2012).
In an even more general picture, erosion occurs at every places where $u_*$ exceeds $u_{*t}$, whether the near-surface wind field displays a convergent or a divergent character. Wind confluence zones result from the large-scale interactions between the topography and near-surface flow. Cold-air drainage currents converge from a large interior area in smaller areas that thus drain the air from a much wider upstream reservoir. Katabatic confluence areas such as Adélie Land are generally characterized by stronger and more persistent winds (Bromwich and Liu, 1996; Parish and Bromwich, 2007), which can thus lead to enhanced snow transport in these regions, erosion where $u_*>u_{*t}$, or in other words, horizontal divergence of the wind transport of snow.
Nevertheless, for clarity I have removed the terms "divergence" and "convergence" from the paragraph you were confused by, and rewritten the sentence as "*In coastal areas, wind redistribution of snow is responsible for an export of mass beyond the ice-sheet margins.*". Note however that Adelie Land is still introduced as a wind confluence area of East Antarctica in the last paragraph of the introduction.

L26-28: "Sublimation of snow particles ..." Please add citations. There are several good papers on drifting snow sublimation out there which deserve citation here.

I originally intended to have this assertion supported by the several papers I refer to at the beginning of the next paragraph but yes, you're entirely right. I have added 3 significant studies (Mann et al., 2000; Bintanja, 2001; Thiery et al., 2012) highlighting and discussing the role of windborne snow sublimation in Antarctica.

L35: "are to be found" should that not be "are found"?

See our response to the next comment.

L36: "in the absolute values attributed to the relative contribution of these various mechanisms." Sounds vague.

I have reformulated the sentence as "Conversely, contrasting results can be found from one study to another in the absolute values attributed to the contribution of wind-driven processes to the large-scale mass transport."

L40: "3 times fewer" seems to refer to "mass fluxes". But either "3 times fewer" refers to occurrence, or it should be "3 times less".

"3 times fewer" does actually refer to "mass fluxes". I have reversed the sentence and written "3 times larger".

L43: "The degree of plausibility of model-dependant features". I assume "model-dependent", but the sentence is a little vague.

I have reformulated the sentence as "model results as well as the assumptions made in the implementation of wind-driven snow physics need to be carefully assessed with independent observations."

L80-82: It reads as if the experiments were only in January 2010, and it should be highlighted here that the dataset is much longer. I would recommend writing something like: "For long term data acquisition following the measurement campaign, two distinct locations ..."

I have changed the two following sentences to account for your suggestions: *"[…] a field campaign specifically dedicated to drifting snow has been initiated […]. Two distinct locations, namely D17 and D47 (Fig. S1), were instrumented for long term-data acquisition and equipped with […]."*

Section 2.1: How is the power supply organized? Can there by a bias in data availability depending on power source? (For example, if the battery tends to be drained towards the end of the winter season).

A power consumption balance has been calculated before installation of each stations according to the whole set of instruments to ensure a continuous power supply throughout the year. Each installation of new instruments (such as for instance, a second 2G-FlowCapt™ sensor and a sonic depth ranger at D17 in late December 2012) came along with installation of new sets of batteries and solar panels to supply for the additional power requirements. The batteries are also checked during each summer visit and replaced when needed. The batteries' voltage is continuously monitored and stored together with the meteorological variables in the datalogger to ensure that the entire system has been designed properly and is sufficiently supplied with energy throughout the winter. Note also that the FlowCapt™ sensors are known to be low-consuming and are moreover continuously solicited by the datalogger (RS232 connection), so we can easily distinguish between instrument failure (absence of response – no data) and data containing null values (absence of drifting snow). Except for the months of May and June at site D47 and a few other instances of malfunction scattered outside of maintenance operations, the dataset is continuous along the measurement periods. I have added the following related lines in the text:

*"The FlowCapt™ is low-power consuming and designed to withstand harsh climate conditions without regular human attendance. At each station battery voltage is monitored and stored together with the meteorological variables in the datalogger to ensure that the entire measurement system is sufficiently supplied with energy throughout the winter. The 2G-FlowCapt™ are continuously solicited by the datalogger (RS232 connection), such that instances of instrument malfunction (absence of response and no data) can be*

*unambiguously distinguished from the absence of drifting snow (data containing null values). A thorough check on the observations was performed and resulted in omission of misleading data wherever necessary. Except for those very few cases, maintenance periods in summer and a major 2-month failure of the lower 2G-FlowCapt™ sensor at D47 in May and June 2012, the dataset is continuous along the respective measurement periods.".*

Section 2.1: It would be good to mention here explicitly that D17 is still operative.
It is now explicitly mentioned in the section and in Table 1.

Section 2.1: Fig. S2 in the supplement should show the dates on which the photos were taken.
Done.

L112: "the drainage of the sinking near-surface air" sounds vague to me.
I assumed that the lack of clarity here came from the use of "sinking" and rewrote the sentence as "The local topography controls the drainage of the dense near-surface air as it flows downslope and accelerates […]".

L113: "over an unobstructed"
Corrected accordingly.

L118: "higher incidence of drifting snow", maybe add a reference to Fig. S4?
Done.

L119: "combined with"
Corrected accordingly.

L118-119: I suggest to reference the accompanying Brief Communication here. "Brief communication: Rare ambient saturation during drifting snow occurrences in coastal East Antarctica"
Done.

L160-161: This seems like an appropriate location to introduce Equation 2, instead of Section 3.4. At least, I assume that the same correction is made when part of the FlowCapt was buried as done in Equation 2?
See my response to general comment #4.

L171: Again, I don't think punctual is appropriate here.
The word "punctual" has been replaced with "sporadic".

L194-195: This statement deserves citations.
I refer now to Schmidt (1980) who discusses the importance of inter-particle bonds in the initiation of snow transport and Amory et al. (2017) who discuss the influence of increased threshold friction velocities in summer on snow mass fluxes from data collected at D17 and relate it to the growth of inter-particle bonds.

Fig. 1, as well as Fig. 2: The figure caption should also mention what the gray shaded areas denote (currently it's only explained in the main text).
I have added in the figure caption the meaning of the shaded areas. Note that CRED distributions have been combined into one figure (see fig. R2 and my response to general comment #3) to facilitate comparison between both locations and the figure is now described according to the two main curves only (comments involving shaded areas have been removed) for clarity.

L268: Maybe add: "h_ref = 2 m, which is the sum of two 1 m long FlowCapt sensors."
Done.

L290: This is confusing. Fig 4, right panel should be a non-log (i.e., linear) scale in order for it to show a linear increase of QT and event duration.

The logarithm scale is preferred here because differences of several orders of magnitude in mass transport per event for all the range of drifting snow events decrease significantly readability when using a linear scale (Fig. R4). I suggest to still make use in the paper of the logarithm scale for readability purposes but include the linear regression fits and their respective equation to highlight the linear character of the relationship between mass transport and duration as shown in Fig. R5.

[Figure]

**Figure R4**. Snow mass transport in drift conditions against duration for each drifting snow event recorded at D47 (red circles) and D17 (blue crosses). Only periods for which two 2G-FlowCapt™ sensors were installed and/or not entirely covered with snow are considered. Linear fits for D47 (black line) and D17 (light blue line) data are also reported on the graph.

[Figure]

**Figure R5**. Logarithm of snow mass transport in drift conditions against duration for each drifting snow event recorded at D47 (red circles) and D17 (blue crosses). Only periods for which two 2G-FlowCapt™ sensors were installed and/or not entirely covered with snow are considered. Linear fits for D47 (black line) and D17 (light blue line) data are also reported on the graph.

L296-297: Could this be substantiated by showing the two levels separately?
Unfortunately the contribution of the saltation and suspension layers to the snow mass flux estimates provided by the 2G-FlowCapt™ cannot be distinguished because fluxes are vertically integrated over the exposed length of the sensor, which for the sensor closest to the ground almost always largely exceeds typical saltation heights (~10 cm) over the measurement period. This has been added to the text.

The author does not distinguish drift from blowing snow phenomena and the threshold of snow sublimation, and their implication on the mass transport/sublimation and the difference between the two sites. Blowing and drifting snow are not redistribution process, a significant part of blowing snow sublimate as pointed out by snow radar survey (see Frezzotti et al., 2007; Eisen et al., 2008) or satellite survey (Scarchilli et al., 2010, Palm et al, 2011, 2017; Scambos et al., 2012). The AWS and FlowCapt sensor provide single measurement point for limited number of years and must be analysed in the contest of Surface Mass Balance study derived from other field measurements as stakes, firn cores, snow radar profile and satellite studies.

Although rather concise, many issues are raised in this report and I'll try to address all of them. Note that some of the elements of response provided here are redundant with the information provided in the manuscript but are reported here since they constitute key elements of the argumentation.

One major issue raised by the reviewer is the lacking character of the publication concerning surface mass balance aspects, and the discrepancy between the reviewer's expectations and the content of the paper. If the subject of the paper were to investigate the relations between erosion and variability in the surface mass balance in Adelie Land, I should have indeed considered those points raised by the reviewer, I agree, although one would face some serious complications by doing so, as I will discuss it in the following paragraph. But the objective of the paper in the proposed version is different: I aim here at publishing and presenting the drifting snow database while providing some examples of use of the data through a first statistical (temporal and spatial) analysis of drifting snow mass transport and frequency (made possible by the high sampling frequency and the continuous character over the respective measurement periods) and in which emphasis is placed on aspects relevant to the modelling of drifting snow. The interest of the paper partly relies on the quality and the open-access character of the drifting snow database (which has been deposited on zenodo and can be now downloaded at https://zenodo.org/record/3630497; see my response to reviewer RC1), that compiles new observations which are almost inexistent in the extreme and remote Antarctic environment. By making them freely available to the scientific community without condition, the paper is an opening to a larger field of

applications, such as evaluation of climate models, simultaneous analysis of ground-based and remotely sensed data, investigations on polar boundary-layer physics or accumulation/ablation processes, each of them belonging to a specific area of expertise and individually warranting a careful, detailed, equally interesting attention.

It is of crucial importance to understand the possibilities offered by the drifting snow data. In particular, the relation between the snow mass flux and ablation at a given area is far from being direct. Snow mass fluxes do not constitute an estimation of local erosion; rather they are the integrated result of all the mechanisms that contribute to the presence, amount and time residence of snow particles in the air, including notably precipitation and advection from upwind areas. Unfortunately the FlowCapt™ sensor does not distinguish neither the source or the geographical origin of the particles impacting the tube, and how much precipitation contribute to the snow mass transport in Antarctica is still an open research question. Therefore these observations must not be perceived as measurements of "wind-driven negative term of the surface mass balance" and cannot be used to quantify local ablation rates without the use of a complementary approach such as numerical modelling. This is the main reason why all the references mentioned by the reviewer are indeed not discussed in the paper since they would lie beyond the scope of the paper. Some of other major concerns that would arise from the multidisciplinary and quite ambitious approach suggested by the reviewer result from the fact that:

- the transect along which stake measurements are indeed performed in Adelie Land is not aligned with the main slope in the wind direction, so any spatial variability depicted in the local SMB signal might not directly correlate to the magnitude of the snow mass flux.

- the fine spatial resolution of the stake networks in Adelie Land demonstrates a high, sub-kilometre spatial variability that couldn't be supported by analysis of mass fluxes performed at only two distant locations, and whose spatial representativeness cannot be assessed in the absence of other comparable measurements in the area.

- the observations presented in this paper are, as you mentioned, "*single measurements point for limited number of years*", moreover performed over the most recent years. A mismatch in timing of several decades can thus be expected with the information contained in deep snow/ice layers sampled through ice cores and radar stratigraphy.

Moreover, temporal and spatial variability of the SMB in Adelie Land has already been quite well investigated (as demonstrated by the long, non-exhaustive list of references mentioned in your report), while drifting snow mass transport has received much less attention. Documentation of spatial et temporal variability in snow mass transport over Antarctica almost exclusively relies on models whose ability to represent drifting snow processes has been extremely limited owing notably to the current extreme scarcity of drifting snow observations. This justifies, in my humble opinion, an initial and independent documentation of the entire database before exploiting further possible connections with other processes and synergetic uses with other products. For the above-mentioned reasons, and because I'm also deeply convinced that keeping the scientific message of a paper as onefold improves clarity, readability and efficiency and thus prevent the paper from being too long with various scientific messages and disconnected sections, I believe that the various applications mentioned by the reviewer are all interesting subjects for separate papers. Finally, note that SMB-related aspects are discussed in a manuscript that I've submitted to Geoscientific Model Development, in which the data presented in this paper have been used in conjunction with the SMB observations in Adelie Land to evaluate the drifting snow scheme of the regional climate model MAR and its ability to represent the variability in accumulation along the transect.

The following comments gather some elements of responses to the remaining comments of the reviewer:

- The driting snow observations are performed far from the blue ice area (i.e. respectively 10 and 100 km away from the ice margin), which covers the very first hundreds of metres of the icesheet margin, and relate to different local (topographical) conditions irrelevant to the objective of the paper in its current form.

- As the sensors are installed at both measurement sites so as to cover the first two meters above ground, which is the height conventionally used to distinguish drifting from blowing snow, they do not enable a distinction between drifting and blowing snow.

- Sublimation of windborne snow is determined by the temperature and humidity gradients across the boundary layer between each snow particle and its environment and is proportional to the undersaturation of the atmosphere (Schmidt, 1982). For a given wind speed, threshold values at which airborne snow sublimation becomes significant can thus be expected to vary significantly depending, among others, on the snow particle concentration and thermodynamic properties and structure of the atmosphere, or the dynamical origin of the boundary-layer flow. From this perspective, as a *"threshold of snow sublimation"* sounds quite vague to me, I also assumed that the reviewer possibly meant "wind speed thresholds for snow transport". A comparison between both measurements sites in terms of occurrence of drifting snow as a function of wind speed is already proposed and discussed in the paper (see Figs. 2 and 3 in the former and revised version of the manuscript, respectively). Moreover, the actual quantity involved in the triggering of drifting snow is the friction velocity, which is only dependent on surface snow properties (Gallée et al., 2001). In the absence of measurements of surface snow properties, and knowing that atmospheric flow conditions would also influence the results, accurate determination of such thresholds can be achieved through turbulence measurements (not available either at D17 or D47) or wind speed profiles using the similarity theory by computing the friction velocity at the onset of drifting snow (e.g., Trouvilliez et al. 2014). However, such an alternative involve a thorough determination protocole (Amory et al., 2017) and selection criteria that are not continuously and homogeneously met at D17 and would result in a discontinuous time series, additionally subject to impeding factors (see my response to general comment #3 of reviewer RC1). For instance, accuracy issues arising from various variable numbers of available anemometers, absence of knowledge of measurement heights, choice of stability correction functions, or the validity of the similarity theory in drifting snow conditions, as well as inclusion of drag effects to the shear stress estimates (e.g., Amory et al., 2016) and artificial roughness created during maintenance operations, are all arguments that would, again, certainly deserve an entire sensitivity study in the form of another publication. Finally, the single measurement level at D47 preclude such determinations, and therefore spatial comparison with D17.

- Limitations in computing sublimation rates from AWS data and drifting snow mass fluxes have already been discussed in a recently published paper (Amory and Kittel 2019). The authors made use of one year of this dataset at site D17 in complement to relative humidity profiles to investigate the development of a near-saturated surface air layer in relation to the occurrence of drifting snow. As also mentioned in that paper, such an exercise involves specific requirements that are only met during a reduced period of time at site D17, justifying its treatment in a separate publication and precluding its application to D47 and outside of the period of study (year 2013). Similarly, strong limitations in the use of the thermo-hygrometers and in the applicability of the Monin-Obukhov similarity theory for retrieving latent heat fluxes at D17 (from which drifting snow sublimation rates could be inferred) have also been discussed in Barral et al. (2014).

- Sublimation of windborne snow has been inferred from accumulation measurements (e.g. Frezzotti et al., 2007; Scambos et al., 2012) but still remained to be confirmed and quantified by measurements of the latent heat flux within the atmosphere and drifting/blowing snow layers, accounting for the physical constraints mentioned above. The usual alternative is the use of gridded model products. Attempts using also satellite data have been made (Palm et al., 2017), but they involve the use of (i) parameterizations for snow particles properties, (ii) snapshots of the atmospheric conditions that are representative of instantaneous conditions only, and (iii) reanalysis produced from model that do not take into account interactions of snow particles with the atmosphere, particularly the negative feedback of windborne snow sublimation thus leading

to a dry bias that can result in strong overestimation of sublimation rates and give the role of an infinite mass sink to the atmosphere. Moreover, as discussed in the introduction, strong discrepancy currently remain between the available model products (~100 vs 400 Gt/an), to the extent that the difference between each estimate is one order of magnitude higher than any other ablation term of the surface mass balance as determined from regional models (e.g., Agosta et al., 2019; Mottram et al., 2020). Different model-based approaches have also been proposed in which drifting snow mass transport is believed to be the first-order process with respect to sublimation, because of the low capacity of the atmosphere to hold moisture in the cold environment where accumulation measurements have been performed (Agosta et al., 2019). The role of sublimation during snow transport, particularly as a negative ablation term in Antarctica, is a currently debated problem in meteorology and snow science. I kindly refer to Amory and Kittel (2019) for a more detailed discussion on that matter from D17 data.

- Note that in Trouvilliez et al. (2014) only the initial results of the drifting snow observation campaign (the first 2 years) are presented under different processing criteria relative to a less complete knowledge on the FlowCapt™ capabilities at the time of redaction, and with no accessibility of the data and much less emphasis on drifting snow mass transport.

- Trouvilliez et al. (2015) focus on data collected in the French Alps and their work is not connected to the observations in Adelie Land.

[revised manuscript text omitted]

**Correspondence:** C. Amory (charles.amory@uliege.be)

**Table S1.** Meteorological instruments installed at D47 and D17 along the respective observation periods (sensors marked with * are manufactured by Campbell Scientific, Inc.). Instrument types and specificities are given for the sensors nearest to 2 m as recovered by the ultrasonic depth gauge (2013-2018) or for the nearest original height when information on surface height is not available (2010-2012).

| | D47 | | | D17 | | |
|---|---|---|---|---|---|---|
| | Period | Sensor | Accuracy | Period | Sensor | Accuracy |
| Wind speed | 01/10 - 12/12 | Young 05103 | $\pm 0.3$ m s$^{-1}$ | 02/12 - 12/10 | RNRG 40C | $\pm 0.14$ m s$^{-1}$ |
| | | | | 01/11 - 12/18 | A100LK* | $\pm 0.1$ m s$^{-1}$ |
| Wind direction | 01/10 - 12/12 | Young 05103 | $\pm 3$ ° | 02/12 - 12/10 | RNRG 200P | $\pm 4$ ° |
| | | | | 12/12 - 12/18 | W200P* | $\pm 2$° |
| Air temperature | 01/10 - 12/12 | Vaisala HMP155 | $\pm 0.3$ °C | 02/10 - 12/18 | Vaisala HMP45 | $\pm 0.4$ °C |
| Relative humidity | 01/10 - 12/12 | Vaisala HMP155 | $\pm 1$ % | 02/10 - 12/18 | Vaisala HMP45 | $\pm 2$% |
| Snow height | 01/10 - 12/12 | SR50A* | $\pm 0.01$ m | 12/12 - 12/18 | SR50A* | $\pm 0.01$ m |
| Snow mass flux | 01/10 - 12/12 | 2G-FlowCapt™ | - | 12/12 - 12/18 | 2G-FlowCapt™ | - |

**Table S2.** Comparison of drifting snow occurrences at D17 and D47 over the period 2010-2012.

| D47 \ D17 | DR | nDR |
|---|---|---|
| DR | 53.7% | 28.3% |
| nDR | 3.4% | 14.6% |

[Figure]

**Figure S1.** Pictures of the meteorological equipment and wind roses at D17 (left panel) and D47 (right panel) for the respective observation periods. At D17 the wind speed from the measurement closest to 2 m is used while wind direction is taken at the upper level of the meteorological mast. The colours indicate the wind speed ranges in m s-1. The pictures were taken in late January 2014 at D17 and in early January 2011 at D47.

[Figure]

**Figure S2.** Monthly timeseries of wind speed (upper panel), temperature (middle panel) and relative humidity (lower panel) at 2-m height for D47 (red circles) and D17 (blue squares) for the respective observation periods 2010-2012 and 2010-2018. Mean values for each variable have been first determined from the measurement level closest to 2 m for each month of the observation period, and averaged within each monthly bin to produce monthly average values.

**S1 Intercomparison between snow particle counters S7 and second-generation FlowCapt™ sensors during a drifting snow event in Adelie Land**

**S1.1 Snow particle counters**

The measurement principle of the snow particle counter S7 (SPC-S7) follows an optical method based on the strong absorption of the infrared light by the snow. The diameter and number flux of snow particles are detected by their shadows on a super-luminescent diode sensor. Electric pulse signals corresponding to a snow particle passing through a sampling area of 50 mm$^2$ (2 mm in height and 25 mm in width) and whose voltage is directly proportional to the size of the particle are classified into 32 size bins from ~40 to 500 $\mu$m (Sato et al., 1993). This means that snow particles smaller than 40 $\mu$m remain undetected and snow particles larger than 500 $\mu$m are assigned to the maximum diameter class. Thanks to a self-steering vane the SPC-S7 measures perpendicularly to the horizontal wind vector the distribution size spectrum of snow particles every 1 s, from which the horizontal snow mass flux, $\eta$, can be computed assuming fully spherical snow particles with a density equal to that of ice as follows

$$\eta = \sum_{id=1}^{32} \eta_d = \sum_{id=1}^{32} n_d \frac{4}{3} \pi \left(\frac{d}{2}\right)^3 \rho_i \qquad (S1)$$

with $\eta_d$ (kg m$^{-2}$ s$^{-1}$) the horizontal snow mass flux for the class of diameter d (m), id the index and $n_d$ the measured number flux of snow particles (part. m$^{-2}$ s$^{-1}$) for each of the 32 diameter classes, and $\rho_i$ the particle density (917 kg m$^{-3}$).

**S1.2 Experimental set-up**

Two SPCs were installed on 28 January 2014 (Fig. S3) a few hours before strong drifting snow occurred in conjunction with strong katabatic winds reinforced by the passage of a low-pressure system off the Adelie Coast. The equipment was removed on 29 January once drifting snow ceased. One SPC was installed at a fixed position 1 m above the ground, while the position of the other was alternatively switched manually between 0.5 and 2 m above the ground every 1-2 hours. This was done in order to study the vertical gradient of the mass flux for two ranges of height (0.1-1.1 m and 1.2-2.2 m) above the snow surface for which 2G-FlowCapt™ measurements are also available for comparison. The high energy requirements of the SPCs (~15 W) were fulfilled by an electric generator that was housed together with the acquisition system in a mobile shelter downwind of the measurement structure. Only a few data are missing due to problem with the acquisition system of the SPC at the beginning of the experiment, resulting in an timeseries almost continuous along the event.

**S1.3 Computation of integrated snow mass fluxes from SPC data**

According to the diffusion theory of drifting snow (Radok, 1977), the averaged drifting snow particle density (kg m$^{-3}$) in the diffusion layer can be approximated by a function of height. When the wind profile follows a power law, an expression for the vertical distribution of the snow mass flux $\eta$(z) (kg m$^{-2}$ s$^{-1}$) writes

$$\eta(z) = az^{-b} \qquad (S2)$$

[Figure]

**Figure S3.** Picture of the snow particle counters installed at D17 during the intercomparison experiment in late January 2014.

where a is the calibration parameter and b the exponent independent of height. These parameters were derived by regression from the data measured by the two SPCs (Trouvilliez et al., 2015), alternatively available for the two height ranges. Then, the half-hourly average of the horizontal snow mass flux vertically integrated over the corresponding height covered by the 2G-FlowCapt™ can be estimated. Because (i) snow depth measurements revealed insignificant height change after the event
35  and were affected by the presence of drifting snow particles perturbing the travel of ultrasound pulses along the measuring path during the event, (ii) the two 2G-Flowcapts were respectively installed at 0.1 and 1.2 m above the snow surface at the beginning of the event, and (iii) the heights of the SPCs were regularly checked and manually adjusted along the experiment, constant heights are used in the integration. Finally, data were processed following the procedure described in Guyomarc'h et al. (2019). Resulting integrated snow mass fluxes are compared in Fig. S4. Although more data are necessary to better assess
40  the performance of the 2G-FlowCaptTM in Antarctic conditions, a high degree of agreement between the two types of sensor is depicted with a correlation coefficient of 0.82 and 0.93 and a rmse of 70 $10^{-3}$ and 13 $10^{-3}$ kg m$^{-2}$ s$^{-1}$ (by taking the SPC-S7 as a reference) for the lower and upper height range, respectively.

[Figure]

**Figure S4.** Comparison between snow mass fluxes provided by 2G-Flowcapt™ sensors and computed from measurements made with snow particle counters (SPC-S7) during a snow transport event at site D17 in late January 2014. A distinction is made between snow mass fluxes integrated over 0.1 to 1.1 m and 1.2 to 2.2 m above ground.

---

## Author Response (AR1)

Dear Louise Sandberg Sørensen,

Thanks you for your response. Please find here the revised version of the manuscript and related updated supplements.

Best regards,
Charles Amory

[revised manuscript text omitted]

---

## Referee Report (RR1)

Reviewer: Stephen Palm

Review of:

**Drifting snow statistics from multiple-year autonomous measurements in Adelie Land, eastern Antarctica**

Charles Amory

Department of Geography, University of Liège, Liège, Belgium
Correspondence: C. Amory (charles.amory@uliege.be)

This paper presents high temporal resolution measurements of drifting and blowing snow from two surface stations in Adelie Land for a 3 and 9 year period. Such measurements are important for understanding the frequency and intensity of drifting and blowing snow and its effect on ice sheet mass balance. Given the paucity of these observations, this work is important and should be published after addressing the minor comments below.

Line 20: Please cite Palm et al., 2018 as this paper was the first to estimate the mass of snow being exported off the Antarctic continent.

Line 41: Change "relatively" to "relative"

Line 42-43: You should change this to at the present time the layer are restricted to 30 m deep, but ICESat-2 will be able to get layers less than 10 m deep. I'm working on that... Or maybe just begin sentence "Moreover, prior satellite detection has been restricted"

Table 1: Indicate relative humidity is respect to ice (or water).

Line 121: Not the right word. Maybe say "This provides evidence that..."

Line 147: Delete the work "occurrences"

Line 168: along is not the right word. "during" is better

Line 176: Change "underestimates" to "underestimate"

Line 177: Change "relatively" to "relative"

Line 180: Change "reached the ground yet is" to "yet reached the ground"

Line 239: Change "evidences" to "demonstrates"

Line 244: Change "evidences" to "shows"

Line 297: "drifting snow events" Shouldn't you say here "significant drifting snow events" since you have filtered them by this criteria?

Line 397: Change "evidence" to "show"

Lines 399-401: I do not understand what you are trying to say here. I suppose it is in reference to satellite lidar's poor spatial sampling and that it cannot sample the same region very frequently. I suggest changing this sentence to:

The poor spatial and temporal coverage of satellite lidar techniques render it difficult to determine the mean duration of snow-transport events such as those reported here. However, blowing snow events covering large areas can be successfully detected and tracked over a period of days as demonstrated in Palm et al, 2011.

---

## Author Response (AR2)

Dear Stephen,

Thank you for your constructive comments. My responses are reported hereafter in red.

Thanks and regards,
Charles

**General comments**

This paper presents high temporal resolution measurements of drifting and blowing snow from two surface stations in Adelie Land for a 3 and 9 year period. Such measurements are important for understanding the frequency and intensity of drifting and blowing snow and its effect on ice sheet mass balance. Given the paucity of these observations, this work is important and should be published after addressing the minor comments below.

Line 20: Please cite Palm et al., 2018 as this paper was the first to estimate the mass of snow being exported off the Antarctic continent.
I think you meant Palm et al. (2017). I've added the reference to the sentence.

Line 41; Change "relatively" to "relative"
Changed accordingly.

Line 42-43: You should change this to at the present time the layer are restricted to 30 m deep, but ICESat-2 will be able to get layers less than 10 m deep. I'm working on that... Or maybe just begin sentence "Moreover, prior satellite detection has been restricted"
Very interesting news for the community. Unfortunately, as this work has been published yet (to my knowledge), I suggest to follow you second recommendation.

Table 1: Indicate relative humidity is respect to ice (or water).
Done.

Line 121: Not the right word. Maybe say "This provides evidence that..."
Changed accordingly.

Line 147: Deletd the word "occurrences"
Changed accordingly.

Line 168: along is not the right word. "during" is better
Changed accordingly.

Line 176: Change "underestimates" to "underestimate"
Changed accordingly.

Line 177: Change "relatively" to "relative"
Changed accordingly.

Line 180: Change "reached the ground yet is" to "yet reached the ground"
Changed accordingly.

Line 239: Change "evidences" to "demonstrates"
Changed accordingly.

Line 244: Change "evidences" to "shows"
Changed accordingly.

Line 297: "drifting snow events" Shouldn't you say here "significant drifting snow events" since you have filtered them by this criteria?
To avoid confusion with the next section in which major drifting snow events are discussed, I'd rather stick to the term "drifting snow events" and speak only of "significant drifting snow event" when mentioning the selection criteria as also applied for sorting out drifting snow events in the Alps (see L272 of the revised manuscript).

Line 397: Change "evidences" to "shows"
Changed accordingly.

Lines 399-401: I do not understand what you are trying to say here. I suppose it is in reference to satellite lidar's poor spatial sampling and that it cannot sample the same region very frequently. I suggest changing this sentence to:

[revised manuscript text omitted]